# Optimization Variance: Exploring Generalization Properties of DNNs

## Abstract

Unlike the conventional wisdom in statistical learning theory, the test error of a deep neural network (DNN) often demonstrates double descent: as the model complexity increases, it first follows a classical U-shaped curve and then shows a second descent. Through bias-variance decomposition, recent studies revealed that the bell-shaped variance is the major cause of model-wise double descent (when the DNN is widened gradually). This paper investigates epoch-wise double descent, i.e., the test error of a DNN also shows double descent as the number of training epochs increases. Specifically, we extend the bias-variance analysis to epoch-wise double descent, and reveal that the variance also contributes the most to the zero-one loss, as in model-wise double descent. Inspired by this result, we propose a novel metric, *optimization variance* (OV), to measure the diversity of model updates caused by the stochastic gradients of random training batches drawn in the same iteration. OV can be estimated using samples from the *training* set only but correlates well with the (unknown) *test* error. It can be used to predict the generalization ability of a DNN when the zero-one loss is used in test, and hence early stopping may be achieved without using a validation set.

## 1 Introduction

Deep Neural Networks (DNNs) usually have large model capacity, but also generalize well. This violates the conventional VC dimension (Vapnik, 1999) or Rademacher complexity theory (Shalev-Shwartz & Ben-David, 2014), inspiring new designs of network architectures (Krizhevsky et al., 2012; Simonyan & Zisserman, 2015; He et al., 2016; Zagoruyko & Komodakis, 2016) and reconsideration of their optimization and generalization (Zhang et al., 2017; Arpit et al., 2017; Wang et al., 2018; Kalimeris et al., 2019; Rahaman et al., 2019; Allen-Zhu et al., 2019).

Model-wise double descent, i.e., as a DNN's model complexity increases, its test error first shows a classical U-shaped curve and then enters a second descent, has been observed on many machine learning models (Advani & Saxe, 2017; Belkin et al., 2019a; Geiger et al., 2019; Maddox et al., 2020; Nakkiran et al., 2020). Multiple studies provided theoretical evidence of this phenomenon in some tractable settings (Mitra, 2019; Hastie et al., 2019; Belkin et al., 2019b; Yang et al., 2020; Bartlett et al., 2020; Muthukumar et al., 2020). Specifically, Neal et al. (2018) and Yang et al. (2020) performed bias-variance decomposition for mean squared error (MSE) and the cross-entropy (CE) loss, and empirically revealed that the bell-shaped curve of the variance is the major cause of model-wise double descent. Maddox et al. (2020) proposed to measure the effective dimensionality of the parameter space, which can be further used to explain model-wise double descent.

Recently, a new double descent phenomenon, epoch-wise double descent, was observed, when increasing the number of training epochs instead of the model complexity[1] (Nakkiran et al., 2020). Compared with model-wise double descent, epoch-wise double descent is relatively less explored. Heckel & Yilmaz (2020) showed that epoch-wise double descent occurs in the situation where different parts of DNNs are learned at different epochs, which can be eliminated by proper scaling of step sizes. Zhang & Wu (2020) discovered that the energy ratio of the high-frequency components of a DNN's prediction landscape, which can reflect the model capacity, switches from increase to

---

[1]In practice, some label noise is often added to the training set to make the epoch-wise double descent more conspicuous.

decrease at a certain training epoch, leading to the second descent of the test error. However, this metric fails to provide further information on generalization, such as the early stopping point, or how the size of a DNN influences its performance.

This paper utilizes bias-variance decomposition of the zero-one (ZO) loss (CE loss is still used in training) to further investigate epoch-wise double descent. By monitoring the behaviors of the bias and the variance, we find that the variance plays an important role in epoch-wise double descent, which dominates and highly correlates with the variation of the test error.

Though the variance correlates well with the test error, estimating its value requires training models on multiple different training sets drawn from the same data distribution, whereas in practice usually only one training set is available[2]. Inspired by the fact that the source of variance comes from the random-sampled training sets, we propose a novel metric, *optimization variance* (OV), to measure the diversity of model updates caused by the stochastic gradients of random training batches drawn in the same iteration. This metric can be estimated from a single model using samples drawn from the *training* set only. More importantly, it correlates well with the *test* error, and thus can be used to determine the early stopping point in DNN training, without using any validation set.

Some complexity measures have been proposed to illustrate the generalization ability of DNNs, such as sharpness (Keskar et al., 2017) and norm-based measures (Neyshabur et al., 2015). However, their values rely heavily on the model parameters, making comparisons across different models very difficult. Dinh et al. (2017) shows that by re-parameterizing a DNN, one can alter the sharpness of its searched local minima without affecting the function it represents; Neyshabur et al. (2018) shows that these measures cannot explain the generalization behaviors when the size of a DNN increases. Our proposed metric, which only requires the logit outputs of a DNN, is less dependent on model parameters, and hence can explain many generalization behaviors, e.g., the test error decreases as the network size increases. Chatterji et al. (2020) proposed a metric called Model Criticality that can explain the superior generalization performance of some architectures over others, yet it is unexplored that whether this metric can be used to indicate generalization in the entire training procedure, especially for some relatively complex generalization behaviors, such as epoch-wise double descent.

To summarize, our contributions are:

- We perform bias-variance decomposition on the test error to explore epoch-wise double descent. We show that the variance dominates the variation of the test classification error.

- We propose a novel metric, OV, which is calculated from the training set only and correlates well with the test classification error.

- Based on the OV, we propose an approach to search for the early stopping point without using a validation set, when the zero-one loss is used in test. Experiments verified its effectiveness.

The remainder of this paper is organized as follows: Section 2 introduces the details of tracing bias and variance over training epochs. Section 3 proposes the OV and demonstrates its ability to indicate the test behaviors. Section 4 draws conclusions and points out some future research directions.

## 2 BIAS AND VARIANCE IN EPOCH-WISE DOUBLE DESCENT

This section presents the details of tracing the bias and the variance during training. We show that the variance dominates the epoch-wise double descent of the test error.

### 2.1 A UNIFIED BIAS-VARIANCE DECOMPOSITION

Bias-variance decomposition is widely used to analyze the generalization properties of machine learning algorithms (Geman et al., 1992; Friedman et al., 2001). It was originally proposed for

---

[2]Assume the training set has $n$ samples. We can partition it into multiple smaller training sets, each with $m$ samples ($m < n$), and then train multiple models. However, the variance estimated from this case would be different from the one estimated from training sets with $n$ samples. We can also bootstrap the original training set into multiple ones, each with $n$ samples. However, the data distribution of each bootstrap replica is different from the original training set, and hence the estimated variance would also be different.

the MSE loss and later extended to other loss functions, e.g., CE and ZO losses (Kong & Dietterich, 1995; Tibshirani, 1996; Kohavi et al., 1996; Heskes, 1998). Our study utilizes a unified bias-variance decomposition that was proposed by Domingos (2000) and applicable to arbitrary loss functions.

Let $(\boldsymbol{x}, \boldsymbol{t})$ be a sample drawn from the data distribution $\mathcal{D}$, where $\boldsymbol{x} \in \mathbb{R}^d$ denotes the $d$-dimensional input, and $\boldsymbol{t} \in \mathbb{R}^c$ the one-hot encoding of the label in $c$ classes. The training set $\mathcal{T} = \{(\boldsymbol{x}_i, \boldsymbol{t}_i)\}_{i=1}^n \sim \mathcal{D}^n$ is utilized to train the model $f : \mathbb{R}^d \rightarrow \mathbb{R}^c$. Let $\boldsymbol{y} = f(\boldsymbol{x}; \mathcal{T}) \in \mathbb{R}^c$ be the probability output of the model $f$ trained on $\mathcal{T}$, and $\mathcal{L}(\boldsymbol{t}, \boldsymbol{y})$ the loss function. The expected loss $\mathbb{E}_\mathcal{T}[\mathcal{L}(\boldsymbol{t}, \boldsymbol{y})]$ should be small to ensure that the model both accurately captures the regularities in its training data, and also generalizes well to unseen data.

According to Domingos (2000), a unified bias-variance decomposition[3] of $\mathbb{E}_\mathcal{T}[\mathcal{L}(\boldsymbol{y}, \boldsymbol{t})]$ is:

$$\mathbb{E}_\mathcal{T}[\mathcal{L}(\boldsymbol{t}, \boldsymbol{y})] = \underbrace{\mathcal{L}(\boldsymbol{t}, \bar{\boldsymbol{y}})}_{\text{Bias}} + \beta \underbrace{\mathbb{E}_\mathcal{T}[\mathcal{L}(\bar{\boldsymbol{y}}, \boldsymbol{y})]}_{\text{Variance}}, \tag{1}$$

where $\beta$ takes different values for different loss functions, and $\bar{\boldsymbol{y}}$ is the expected output:

$$\bar{\boldsymbol{y}} = \underset{\boldsymbol{y}^* \in \mathbb{R}^c \mid \sum_{k=1}^c y_k^* = 1, y_k^* \geq 0}{\arg\min} \mathbb{E}_\mathcal{T}[\mathcal{L}(\boldsymbol{y}^*, \boldsymbol{y})]. \tag{2}$$

$\bar{\boldsymbol{y}}$ minimizes the variance term in (1), which can be regarded as the "center" or "ensemble" of $\boldsymbol{y}$ w.r.t. different $\mathcal{T}$.

Table 1 shows specific forms of $\mathcal{L}$, $\bar{\boldsymbol{y}}$, and $\beta$ for different loss functions (the detailed derivations can be found in Appendix A). This paper focuses on the bias-variance decomposition of the ZO loss, because epoch-wise double descent of the test error is more obvious when the ZO loss is used (see Appendix C). To capture the overall bias and variance, we analyzed $\mathbb{E}_{\boldsymbol{x}, \boldsymbol{t}} \mathbb{E}_\mathcal{T}[\mathcal{L}(\boldsymbol{t}, \boldsymbol{y})]$, i.e., the expectation of $\mathbb{E}_\mathcal{T}[\mathcal{L}(\boldsymbol{t}, \boldsymbol{y})]$ over the distribution $\mathcal{D}$.

Table 1: Bias-variance decomposition for different loss functions. The CE loss herein is the complete form of the commonly used one, originated from the Kullback-Leibler divergence. $Z = \sum_{k=1}^c \exp\{\mathbb{E}_\mathcal{T}[\log y_k]\}$ is a normalization constant independent of $k$. $\text{H}(\cdot)$ is the hard-max which sets the maximal element to 1 and others to 0. $\mathbf{1}_{\text{con}}\{\cdot\}$ is an indicator function which equals 1 if its argument is true, and 0 otherwise. $\log$ and $\exp$ are element-wise operators.

| Loss | $\mathcal{L}(\boldsymbol{t}, \boldsymbol{y})$ | $\bar{\boldsymbol{y}}$ | $\beta$ |
|------|------|------|------|
| MSE | $\|\boldsymbol{t} - \boldsymbol{y}\|_2^2$ | $\mathbb{E}_\mathcal{T}\boldsymbol{y}$ | 1 |
| CE | $\sum_{k=1}^c t_k \log \frac{t_k}{y_k}$ | $\frac{1}{Z}\exp\{\mathbb{E}_\mathcal{T}[\log \boldsymbol{y}]\}$ | 1 |
| ZO | $\mathbf{1}_{\text{con}}\{\text{H}(\boldsymbol{t}) \neq \text{H}(\boldsymbol{y})\}$ | $\text{H}(\mathbb{E}_\mathcal{T}[\text{H}(\boldsymbol{y})])$ | 1 if $\bar{\boldsymbol{y}} = \boldsymbol{t}$, otherwise $-P_\mathcal{T}(\text{H}(\boldsymbol{y}) = \boldsymbol{t} \mid \bar{\boldsymbol{y}} \neq \text{H}(\boldsymbol{y}))$ |

## 2.2 TRACE THE BIAS AND VARIANCE TERMS OVER TRAINING EPOCHS

To trace the bias term $\mathbb{E}_{\boldsymbol{x}, \boldsymbol{t}}[\mathcal{L}(\boldsymbol{t}, \bar{\boldsymbol{y}})]$ and the variance term $\mathbb{E}_{\boldsymbol{x}, \boldsymbol{t}} \mathbb{E}_\mathcal{T}[\mathcal{L}(\bar{\boldsymbol{y}}, \boldsymbol{y})]$ w.r.t. the training epoch, we need to sample several training sets and train models on them respectively, so that the bias and variance terms can be estimated from them.

Concretely, let $\mathcal{T}^*$ denote the test set, $f(\boldsymbol{x}; \mathcal{T}_j, q)$ the model $f$ trained on $\mathcal{T}_j \sim \mathcal{D}^n$ $(j = 1, 2, ..., K)$ for $q$ epochs. Then, the estimated bias and variance terms at the $q$-th epoch, denoted as $B(q)$ and $V(q)$, respectively, can be written as:

$$B(q) = \mathbb{E}_{(\boldsymbol{x}, \boldsymbol{t}) \in \mathcal{T}^*}\left[\mathcal{L}\left(\boldsymbol{t}, \bar{f}(\boldsymbol{x}; q)\right)\right], \tag{3}$$

$$V(q) = \mathbb{E}_{(\boldsymbol{x}, \boldsymbol{t}) \in \mathcal{T}^*}\left[\frac{1}{K}\sum_{j=1}^K \mathcal{L}(\bar{f}(\boldsymbol{x}; q), f(\boldsymbol{x}; \mathcal{T}_j, q))\right], \tag{4}$$

---

[3]In real-world situations, the expected loss consists of three terms: bias, variance, and noise. Similar to Yang et al. (2020), we view $\boldsymbol{t}$ as the groundtruth and ignore the noise term.

where

$$\bar{f}(\boldsymbol{x}; q) = \text{H}\left(\sum_{j=1}^{K} \text{H}(f(\boldsymbol{x}; \mathcal{T}_j, q))\right), \tag{5}$$

is the voting result of $\{f(\boldsymbol{x}; \mathcal{T}_j, q)\}_{j=1}^{K}$.

We should emphasize that, in real-world situations, $\mathcal{D}$ cannot be obtained, hence $\mathcal{T}_j$ in our experiments was randomly sampled from the training set (we sampled 50% training data for each $\mathcal{T}_j$). As a result, despite of showing the cause of epoch-wise double descent, the behaviors of bias and variance may be different when the whole training set is used.

We considered ResNet (He et al., 2016) and VGG (Simonyan & Zisserman, 2015) models[4] trained on SVHN (Netzer et al., 2011), CIFAR10 (Krizhevsky, 2009), and CIFAR100 (Krizhevsky, 2009). SGD and Adam (Kingma & Ba, 2014) optimizers with different learning rates were used. The batchsize was set to 128, and all models were trained for 250 epochs with data augmentation. Prior to sampling $\{\mathcal{T}_j\}_{j=1}^{K}$ ($K = 5$) from the training set, 20% labels of the training data were randomly shuffled to introduce epoch-wise double descent.

Figure 1 shows the expected ZO loss and its bias and variance. The bias descends rapidly at first and then generally converges to a low value, whereas the variance behaves almost exactly the same as the test error, mimicking even small fluctuations of the test error. To stabilize that, we performed additional experiments with different optimizers, learning rates, and levels of label noise (see Appendices E and G). All experimental results demonstrated that it is mainly the variance that contributes to epoch-wise double descent.

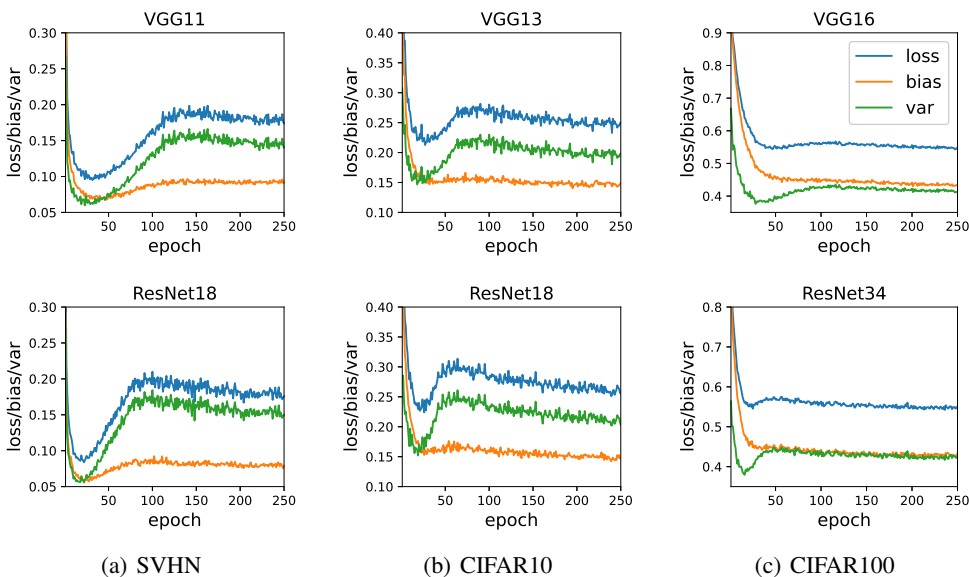

Figure 1: The expected test ZO loss and its bias and variance. The models were trained with 20% label noise. Adam optimizer with learning rate 0.0001 was used.

## 2.3 DISCUSSION

Contradicting to the traditional view that the variance keeps increasing because of overfitting, our experimental results show a more complex behavior: the variance starts high and then decreases rapidly, followed by a bell curve. The difference at the beginning (when the number of epochs is small) is mainly due to the choice of loss functions (see experimental results of bias-variance decomposition for MSE and CE losses in Appendix F). CE and MSE losses, analyzed in the traditional

---

[4]Adapted from https://github.com/kuangliu/pytorch-cifar

learning theory, can reflect the degree of difference of probabilities, whereas the ZO loss only the labels. At the early stage of training, the output probabilities are close to random guesses, and hence a small difference in probabilities may lead to completely different labels, resulting in the distinct variance for different loss functions. However, the reason why the variance begins to diminish at the late phase of training is still unclear. We will explore this problem in our future research.

## 3 Optimization Variance (OV)

This section proposes a new metric, OV, to measure the diversity of model updates introduced by random training batches during optimization. This metric can indicate test behaviors without any validation set.

### 3.1 Notation and Definition

Section 2 verified the synchronization between the test error and the variance, but its application is limited because estimating the variance requires: 1) a test set, and, 2) models trained on different training sets drawn from the same data distribution. It'd be desirable to capture the test behavior of a DNN using a single training set only, without a test set.

According to the definition in (1), the variance measures the model diversity caused by different training samples drawn from the same distribution, i.e., the outputs of DNN change according to the sampled training set. As the gradients are usually the only information transferred from training sets to models during the optimization of DNN, we need to measure the variance of a DNN introduced by the gradients calculated from different training batches. More specifically, we'd like to develop a metric to reflect the function robustness of DNNs to sampling noises. If the function captured by a DNN drastically varies w.r.t. different training batches, its generalization error is very likely to be poor due to a large variance introduced by the optimization procedure. A similar metric is the sharpness of local minima proposed by Keskar et al. (2017), which measures the robustness of local minima as an indicator of the generalization error. However, this metric is only meaningful for local minima and hence cannot be applied in the entire optimization process.

Mathematically, for a sample $(\boldsymbol{x}, \boldsymbol{t}) \sim \mathcal{D}$, let $f(\boldsymbol{x}; \boldsymbol{\theta})$ be the logit output of a DNN with parameter $\boldsymbol{\theta}$. Let $\mathcal{T}_B \sim \mathcal{D}^m$ be a training batch with $m$ samples, $g : \mathcal{T}_B \to \mathbb{R}^{|\boldsymbol{\theta}|}$ the optimizer outputting the update of $\boldsymbol{\theta}$ based on $\mathcal{T}_B$. Then, we can get the function distribution $F_{\boldsymbol{x}}(\mathcal{T}_B)$ over a training batch $\mathcal{T}_B$, i.e., $f(\boldsymbol{x}; \boldsymbol{\theta} + g(\mathcal{T}_B)) \sim F_{\boldsymbol{x}}(\mathcal{T}_B)$. The variance of $F_{\boldsymbol{x}}(\mathcal{T}_B)$ reflects the model diversity caused by different training batches. The formal definition of OV is given below.

**Definition 1 (Optimization Variance (OV))** *Given an input $\boldsymbol{x}$ and model parameters $\boldsymbol{\theta}_q$ at the $q$-th training epoch, the OV on $\boldsymbol{x}$ at the $q$-th epoch is defined as*

$$OV_q(\boldsymbol{x}) \triangleq \frac{\mathbb{E}_{\mathcal{T}_B} \left[ \| f(\boldsymbol{x}; \boldsymbol{\theta}_q + g(\mathcal{T}_B)) - \mathbb{E}_{\mathcal{T}_B} f(\boldsymbol{x}; \boldsymbol{\theta}_q + g(\mathcal{T}_B)) \|_2^2 \right]}{\mathbb{E}_{\mathcal{T}_B} \left[ \| f(\boldsymbol{x}; \boldsymbol{\theta}_q + g(\mathcal{T}_B)) \|_2^2 \right]}. \tag{6}$$

Note that $OV_q(\boldsymbol{x})$ measures the relative variance, because the denominator in (6) eliminates the influence of the logit's norm. In this way, $OV_q(\boldsymbol{x})$ at different training phases can be compared. The motivation here comes from the definition of coefficient of variation[5] (CV) in probability theory and statistics, which is also known as the relative standard deviation. CV is defined as the ratio between the standard deviation and the mean, and is independent of the unit in which the measurement is taken. Therefore, CV enables comparing the relative diversity between two different measurements.

In terms of OV, the variance of logits, i.e., the numerator of OV, is not comparable across epochs due to the influence of their norm. In fact, even if the variance of logits maintains the same during the whole optimization process, its influence on the decision boundary is limited when the logits are large. Consequently, by treating the norm of logits as the measurement unit, following CV we set OV to $\sum_i \sigma_i^2 / \sum_i \mu_i^2$, where $\sigma_i$ and $\mu_i$ represent the standard deviation and the mean of the $i$-th logit, respectively. If we remove the denominator, the value of OV will no longer has the indication ability for generalization error, especially at the early stage of the optimization process.

---

[5]https://en.wikipedia.org/wiki/Coefficient_of_variation

Intuitively, the OV represents the inconsistency of gradients' influence on the model. If $OV_q(\boldsymbol{x})$ is very large, then the models trained with different sampled $\mathcal{T}_B$ may have distinct outputs for the same input, leading to high model diversity and hence large variance. Note that here we emphasize the inconsistency of model updates rather than the gradients themselves. The latter can be measured by the gradient variance. The gradient variance and the OV are different, because sometimes diverse gradients may lead to similar changes of the function represented by DNN, and hence small OV. More on the relationship between the two variances can be found in Appendix B.

### 3.2 EXPERIMENTAL RESULTS

We calculated the expectation of the OV over $\boldsymbol{x}$, i.e., $\mathbb{E}_{\boldsymbol{x}}[OV_q(\boldsymbol{x})]$, which was estimated from 1,000 random training samples. The test set was not involved at all.

Figure 2 shows how the test accuracy (solid curves) and $\mathbb{E}_{\boldsymbol{x}}[OV_q(\boldsymbol{x})]$ (dashed curves) change with the number of training epochs. Though sometimes the OV may not exhibit clear epoch-wise double descent, e.g., VGG16 in Figure 2(c), the symmetry between the solid and dashed curves generally exist, suggesting that the OV, which is calculated from the training set only, is capable of predicting the variation of the test accuracy. Similar results can also be observed using different optimizers and learning rates (see Appendix I).

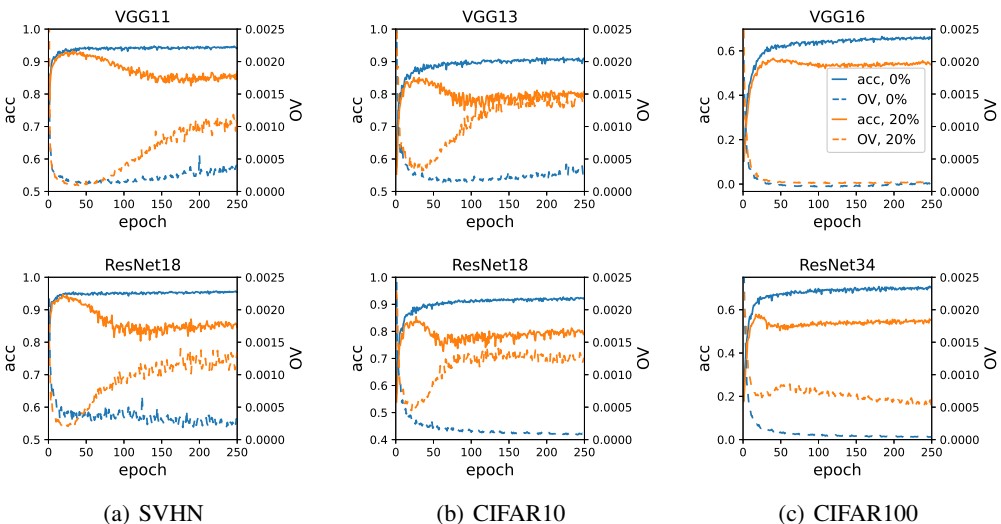

Figure 2: Test accuracy and OV. The models were trained with Adam optimizer (learning rate 0.0001). The number in each legend indicates its percentage of label noise.

Note that epoch-wise double descent is not a necessary condition for applying OV. As shown in Figure 2 (blue curves), we compared the values of OV and generalization errors of DNNs when there are 0% label noises, from which we can see the curves of generalization errors have no epoch-wise double descent, yet the proposed OV still works pretty well.

$OV_q(\boldsymbol{x})$ in Figure 2 was estimated on all training batches; however, this may not be necessary: a small number of training batches are usually enough. To demonstrate this, we trained ResNet and VGG on several datasets using Adam optimizer with learning rate 0.0001, and estimated $OV_q(\boldsymbol{x})$ from different number of training batches. The results in Figure 3 show that we can well estimate the OV using as few as 10 training batches.

Another intriguing finding is that even unstable variations of the test accuracy can be reflected by the OV. This correspondence is clearer on simpler datasets, e.g., MNIST (LeCun et al., 1998) and FashionMNIST (Xiao et al., 2017). Figure 4 shows the test accuracy and OV for LeNet-5 (LeCun et al., 1998) trained on MNIST and FashionMNIST without label noise. Spikes of the OV and the test accuracy happen simultaneously at the same epoch.

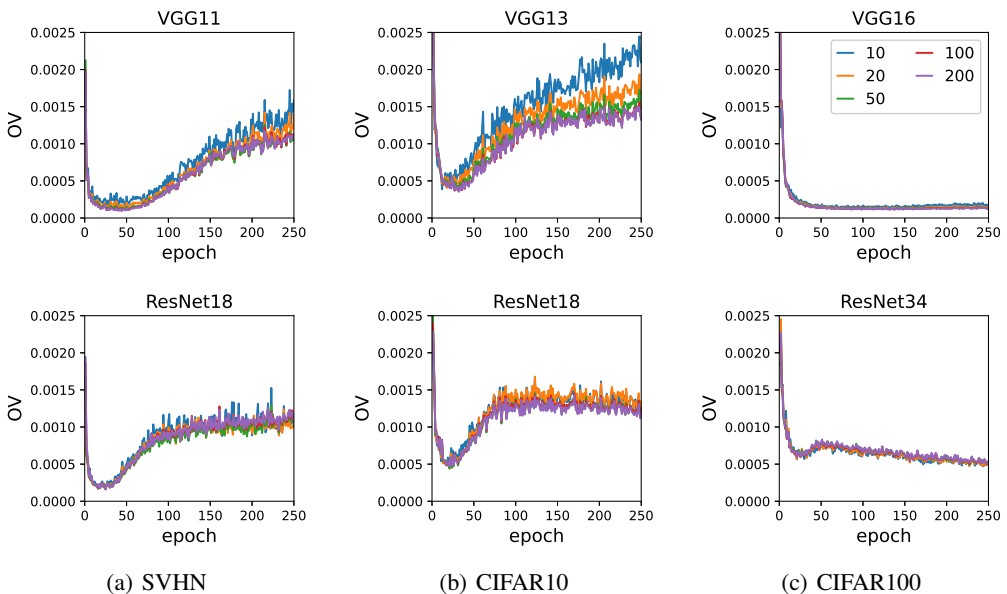

Figure 3: OV estimated from different number of training batches. The models were trained with 20% label noise. Adam optimizer with learning rate 0.0001 was used.

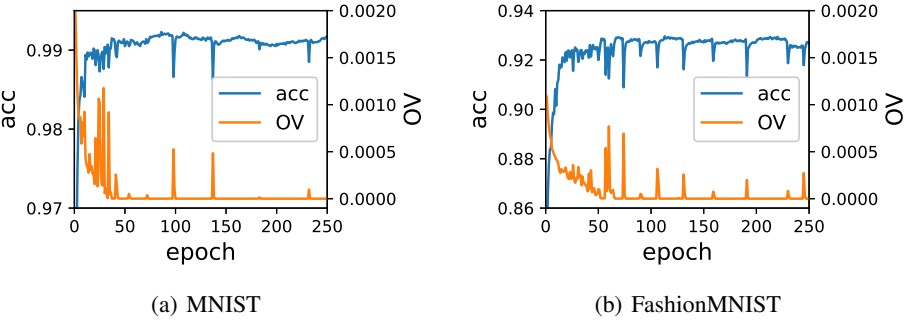

Figure 4: Test accuracy and OV. The model was LeNet-5 trained on MNIST and FashionMNIST with Adam optimizer (learning rate 0.0001).

Our experimental results demonstrate that the generalization ability of a DNN can be indicated by the OV during stochastic training, without using a validation set. This phenomenon can be used to determine the early stopping point, and beyond.

### 3.3 EARLY STOPPING WITHOUT A VALIDATION SET

The common process to train a DNN involves three steps: 1) partition the dataset into a training set and a validation set; 2) use the training set to optimize the DNN parameters, and the validation set to determine when to stop training, i.e., early stopping, and record the early stopping point; 3) train the DNN on the entire dataset (combination of training and validation sets) for the same number of epochs. However, there is no guarantee that the early stopping point on the training set is the same as the one on the entire dataset. So, an interesting questions is: is it possible to directly perform early stopping on the entire dataset, without a validation set?

The OV can be used for this purpose. For more robust performance, instead of using the OV directly, we may need to smooth it to alleviate random fluctuations.

As an example, we smoothed the OV by a moving average filter of 10 epochs, and then performed early stopping on the smoothed OV with a patience of 10 epochs. As a reference, early stopping with the same patience was also performed directly on the test accuracy to get the groundtruth. However, it should be noted that the latter is unknown in real-world applications. It is provided for verification purpose only.

We trained different DNN models on several datasets (SVHN: VGG11 and ResNet18; CIFAR10: VGG13 and ResNet18; CIFAR100: VGG16 and ResNet34) with different levels of label noise (10% and 20%) and optimizers (Adam with learning rate 0.001 and 0.0001, SGD with momentum 0.9 and learning rate 0.01 and 0.001). Then, we compared the groundtruth early stopping point and the test accuracy with those found by performing early stopping on the OV[6]. The results are shown in Figure 5. The true early stopping points and those found from the OV curve were generally close, though there were some exceptions, e.g., the point near (40, 100) in Figure 5(a). However, the test errors, which are what a model designer really cares about, were always close.

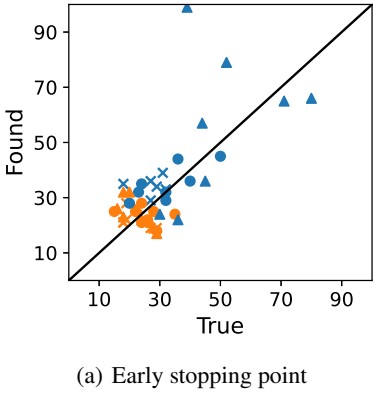

(a) Early stopping point

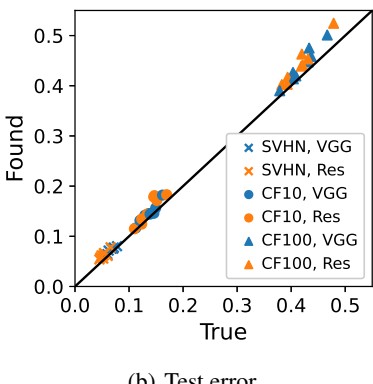

(b) Test error

Figure 5: Early stopping based on test error (True) and the corresponding OV (Found). The shapes represent different datasets, whereas the colors indicate different categories of DNNs ("CF" and "Res" are short for "CIFAR" and "ResNet", respectively).

### 3.4 SMALL SIZE OF THE TRAINING SET

For large datasets, a validation set can be easily partitioned from the training set without hurting the generalization performance. Therefore, OV is more useful in the case of small datasets. To this end, we performed experiments with small numbers (2000, 4000, 6000) of the training samples in CIFAR10 to verify the effectiveness of OV in this situation. Considering the limited numbers of training samples, we trained a small Convolution Neural Network (CNN) using Adam optimizer with learning rate 0.0001, whose detailed information can be found in Appendix H.

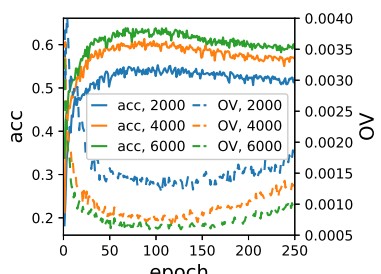

Figure 6: Test accuracy and OV using small sizes of training sets. The numbers represent how many training samples are used for training.

The experimental results are shown in Figure 6. It can be observed that: a) When the size of the training set is small, OV still correlates well with the generalization performance as a function of the training epochs, which verifies the validity of our results on small datasets; b) As expected, more training samples usually lead to better generalization performance, which can also be reflected by comparing the values of OV.

---

[6]Training VGG11 on SVHN with Adam optimizer and learning rate 0.001 was unstable (see Appendix D), so we did not include its results in Figure 5.

### 3.5 Network Size

In addition to indicating the early stopping point, the OV can also explain some other generalization behaviors, such as the influence of the network size. To verify that, we trained ResNet18 with different network sizes on CIFAR10 for 100 epochs with no label noise, using Adam optimizer with learning rate 0.0001. For each convolutional layer, we set the number of filters $k/4$ ($k = 1, 2, ..., 8$) times the number of filters in the original model. We then examined the OV of ResNet18 with different network sizes to validate its correlation with the test accuracy. Note that we used SGD optimizer with learning rate 0.001 and no momentum to calculate the OV, so that the cumulative influence during training can be removed to make the comparison more fair.

The results are shown in Figure 7. As $k$ increases, the OV gradually decreases, i.e., the diversity of model updates introduced by different training batches decreases when widening ResNet18, suggesting that increasing the network size can improve the model's resilience to sampling noise, which leads to better generalization performance. The Pearson correlation coefficient between the OV and the test accuracy reached $-0.94$ ($p = 0.0006$).

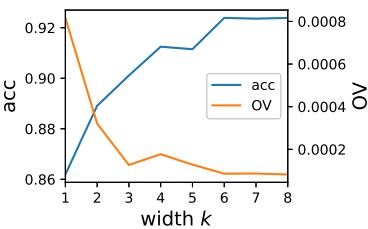

Figure 7: Test accuracy and OV w.r.t. the network size.

Lastly, we need to point out that we did not observe a strong cross-model correlation between the OV and the test accuracy when comparing the generalization ability of significantly different model architectures, e.g., VGG and ResNet. Our future research will look for a more universal cross-model metric to illustrate the generalization performance.

## 4 Conclusions

This paper has shown that the variance dominates the epoch-wise double descent, and highly correlates with the test error. Inspired by this finding, we proposed a novel metric called optimization variance, which is calculated from the training set only but powerful enough to predict how the test error changes during training. Based on this metric, we further proposed an approach to perform early stopping without any validation set. Remarkably, we demonstrated that the training set itself may be enough to predict the generalization ability of a DNN, without a dedicated validation set.

Our future work will: 1) apply the OV to other tasks, such as regression problems, unsupervised learning, and so on; 2) figure out the cause of the second descent of the OV; and, 3) design regularization approaches to penalize the OV for better generalization performance.

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

## A    BIAS-VARIANCE DECOMPOSITION FOR DIFFERENT LOSS FUNCTIONS

This section presents detailed deduction of bias-variance decomposition for different loss functions.

### A.1    THE MEAN SQUARED ERROR (MSE) LOSS

For the MSE loss, we have $\mathcal{L}(\boldsymbol{t}, \boldsymbol{y}) = \|\boldsymbol{t} - \boldsymbol{y}\|_2^2$, and need to calculate $\bar{\boldsymbol{y}}$ based on (2). We first ignore the constraints and solve the following problem:

$$\widetilde{\boldsymbol{y}} = \underset{\boldsymbol{y}^*}{\arg\min}\, \mathbb{E}_{\mathcal{T}}[\|\boldsymbol{y}^* - \boldsymbol{y}\|_2^2], \tag{7}$$

whose solution is $\widetilde{\boldsymbol{y}} = \mathbb{E}_{\mathcal{T}}\boldsymbol{y}$. It can be easily verified that $\widetilde{\boldsymbol{y}}$ satisfies the constraints in (2), and hence $\bar{\boldsymbol{y}} = \widetilde{\boldsymbol{y}} = \mathbb{E}_{\mathcal{T}}\boldsymbol{y}$.

Then, we can decompose the MSE loss as:

$$\begin{aligned} \mathbb{E}_{\mathcal{T}}[\|\boldsymbol{t} - \boldsymbol{y}\|_2^2] &= \mathbb{E}_{\mathcal{T}}[\|\boldsymbol{t} - \bar{\boldsymbol{y}} + \bar{\boldsymbol{y}} - \boldsymbol{y}\|_2^2] \\ &= \mathbb{E}_{\mathcal{T}}[\|\boldsymbol{t} - \bar{\boldsymbol{y}}\|_2^2 + \|\bar{\boldsymbol{y}} - \boldsymbol{y}\|_2^2 + 2(\boldsymbol{t} - \bar{\boldsymbol{y}})^T(\bar{\boldsymbol{y}} - \boldsymbol{y})] \\ &= \|\boldsymbol{t} - \bar{\boldsymbol{y}}\|_2^2 + \mathbb{E}_{\mathcal{T}}[\|\bar{\boldsymbol{y}} - \boldsymbol{y}\|_2^2] + 0, \end{aligned} \tag{8}$$

where the first term denotes the bias, and the second denotes the variance. We can also get $\beta = 1$.

### A.2    CROSS-ENTROPY (CE) LOSS

For $\mathcal{L}(\boldsymbol{t}, \boldsymbol{y}) = \sum_{k=1}^{c} t_k \log \frac{t_k}{y_k}$, $\bar{\boldsymbol{y}}$ can be obtained by applying the Lagrange multiplier method (Boyd & Vandenberghe, 2004) to (2):

$$l(\boldsymbol{y}^*, \lambda) = \mathbb{E}_{\mathcal{T}}\left[\sum_{k=1}^{c} y_k^* \log \frac{y_k^*}{y_k}\right] + \lambda \cdot \left(1 - \sum_{k=1}^{c} y_k^*\right). \tag{9}$$

To minimize $l(\boldsymbol{y}^*, \lambda)$, we need to compute its partial derivatives to $\boldsymbol{y}^*$ and $\lambda$:

$$\frac{\partial l}{\partial y_k^*} = \mathbb{E}_{\mathcal{T}}\left[\log \frac{y_k^*}{y_k} + 1\right] - \lambda, \quad k = 1, 2, ..., c$$

$$\frac{\partial l}{\partial \lambda} = 1 - \sum_{k=1}^{c} y_k^*.$$

By setting these derivatives to 0, we get:

$$\bar{y}_k = \frac{1}{Z} \exp\{\mathbb{E}_{\mathcal{T}}[\log y_k]\}, \quad k = 1, 2, ..., c \tag{10}$$

where $Z = \sum_{k=1}^{c} \exp\{\mathbb{E}_{\mathcal{T}}[\log y_k]\}$ is a normalization constant independent of $k$.

Because

$$\mathbb{E}_{\mathcal{T}}\left[\sum_{k=1}^{c} \alpha_k \log \frac{\bar{y}_k}{y_k}\right] = -\log Z, \quad \forall_{\alpha_k} \sum_{k=1}^{c} \alpha_k = 1, \tag{11}$$

we have:

$$\begin{aligned} \mathbb{E}_{\mathcal{T}}\left[\sum_{k=1}^{c} t_k \log \frac{t_k}{y_k}\right] &= \mathbb{E}_{\mathcal{T}}\left[\sum_{k=1}^{c} t_k \left(\log \frac{t_k}{\bar{y}_k} + \log \frac{\bar{y}_k}{y_k}\right)\right] \\ &= \sum_{k=1}^{c} t_k \log \frac{t_k}{\bar{y}_k} + \mathbb{E}_{\mathcal{T}}\left[\sum_{k=1}^{c} t_k \log \frac{\bar{y}_k}{y_k}\right] \\ &= \sum_{k=1}^{c} t_k \log \frac{t_k}{\bar{y}_k} - \log Z \\ &= \sum_{k=1}^{c} t_k \log \frac{t_k}{\bar{y}_k} + \mathbb{E}_{\mathcal{T}}\left[\sum_{k=1}^{c} \bar{y}_k \log \frac{\bar{y}_k}{y_k}\right], \end{aligned} \tag{12}$$

from which we obtain $\beta = 1$.

### A.3 THE ZERO-ONE (ZO) LOSS

For the ZO loss, i.e., $\mathcal{L}(\boldsymbol{t}, \boldsymbol{y}) = \mathbf{1}_{\text{con}}\{\text{H}(\boldsymbol{t}) \neq \text{H}(\boldsymbol{y})\}$, $\bar{\boldsymbol{y}}$ is the voting result, i.e., $\text{H}(\mathbb{E}_{\mathcal{T}}[\text{H}(\boldsymbol{y})])$, so that the variance can be minimized. However, the value of $\beta$ depends on the relationship between $\bar{\boldsymbol{y}}$ and $\boldsymbol{t}$.

When $\bar{\boldsymbol{y}} = \boldsymbol{t}$, we have:

$$
\begin{aligned}
\mathbb{E}_{\mathcal{T}}\left[\mathbf{1}_{\text{con}}\{\text{H}(\boldsymbol{t}) \neq \text{H}(\boldsymbol{y})\}\right] &= 0 + \mathbb{E}_{\mathcal{T}}\left[\mathbf{1}_{\text{con}}\{\text{H}(\bar{\boldsymbol{y}}) \neq \text{H}(\boldsymbol{y})\}\right] \\
&= \mathbf{1}_{\text{con}}\{\text{H}(\boldsymbol{t}) \neq \text{H}(\bar{\boldsymbol{y}})\} + \mathbb{E}_{\mathcal{T}}\left[\mathbf{1}_{\text{con}}\{\text{H}(\bar{\boldsymbol{y}}) \neq \text{H}(\boldsymbol{y})\}\right],
\end{aligned} \tag{13}
$$

clearly, $\beta = 1$.

When $\bar{\boldsymbol{y}} \neq \boldsymbol{t}$, we have:

$$
\begin{aligned}
\mathbb{E}_{\mathcal{T}}\left[\mathbf{1}_{\text{con}}\{\text{H}(\boldsymbol{t}) \neq \text{H}(\boldsymbol{y})\}\right] &= P_{\mathcal{T}}\left(\text{H}(\boldsymbol{y}) \neq \boldsymbol{t}\right) = 1 - P_{\mathcal{T}}\left(\text{H}(\boldsymbol{y}) = \boldsymbol{t}\right) \\
&= \mathbf{1}_{\text{con}}\{\text{H}(\boldsymbol{t}) \neq \text{H}(\bar{\boldsymbol{y}})\} \\
&\quad - P_{\mathcal{T}}\left(\text{H}(\boldsymbol{y}) = \boldsymbol{t}\middle|\text{H}(\boldsymbol{y}) = \bar{\boldsymbol{y}}\right) P_{\mathcal{T}}\left(\text{H}(\boldsymbol{y}) = \bar{\boldsymbol{y}}\right) \\
&\quad - P_{\mathcal{T}}\left(\text{H}(\boldsymbol{y}) = \boldsymbol{t}\middle|\text{H}(\boldsymbol{y}) \neq \bar{\boldsymbol{y}}\right) P_{\mathcal{T}}\left(\text{H}(\boldsymbol{y}) \neq \bar{\boldsymbol{y}}\right).
\end{aligned} \tag{14}
$$

Since $\bar{\boldsymbol{y}} \neq \text{H}(\boldsymbol{t})$, it follows that

$$
P_{\mathcal{T}}\left(\text{H}(\boldsymbol{y}) = \boldsymbol{t}\middle|\text{H}(\boldsymbol{y}) = \bar{\boldsymbol{y}}\right) = 0. \tag{15}
$$

Then, (14) becomes:

$$
\begin{aligned}
\mathbb{E}_{\mathcal{T}}\left[\mathbf{1}_{\text{con}}\{\text{H}(\boldsymbol{t}) \neq \text{H}(\boldsymbol{y})\}\right] &= \mathbf{1}_{\text{con}}\{\text{H}(\boldsymbol{t}) \neq \text{H}(\bar{\boldsymbol{y}})\} - P_{\mathcal{T}}\left(\text{H}(\boldsymbol{y}) = \boldsymbol{t}\middle|\text{H}(\boldsymbol{y}) \neq \bar{\boldsymbol{y}}\right) P_{\mathcal{T}}\left(\text{H}(\boldsymbol{y}) \neq \bar{\boldsymbol{y}}\right) \\
&= \mathbf{1}_{\text{con}}\{\text{H}(\boldsymbol{t}) \neq \text{H}(\bar{\boldsymbol{y}})\} \\
&\quad - P_{\mathcal{T}}\left(\text{H}(\boldsymbol{y}) = \boldsymbol{t}\middle|\text{H}(\boldsymbol{y}) \neq \bar{\boldsymbol{y}}\right) \mathbb{E}_{\mathcal{T}}\left[\mathbf{1}_{\text{con}}\{\text{H}(\bar{\boldsymbol{y}}) \neq \text{H}(\boldsymbol{y})\}\right],
\end{aligned} \tag{16}
$$

hence, $\beta = -P_{\mathcal{T}}\left(\text{H}(\boldsymbol{y}) = \boldsymbol{t}\middle|\text{H}(\boldsymbol{y}) \neq \bar{\boldsymbol{y}}\right)$.

## B CONNECTIONS BETWEEN THE OPTIMIZATION VARIANCE AND THE GRADIENT VARIANCE

This section shows the connection between the gradient variance and the optimization variance in Definition 1.

For simplicity, we ignore $q$ in $OV_q(\boldsymbol{x})$ and denote $g(\mathcal{T}_B) - \mathbb{E}_{\mathcal{T}_B} g(\mathcal{T}_B)$ by $\tilde{g}(\mathcal{T}_B)$. Then, the gradient variance $V_g$ can be written as:

$$
V_g = \mathbb{E}_{\mathcal{T}_B}\left[\|g(\mathcal{T}_B) - \mathbb{E}_{\mathcal{T}_B} g(\mathcal{T}_B)\|_2^2\right] = \mathbb{E}_{\mathcal{T}_B}\left[\tilde{g}(\mathcal{T}_B)^T \tilde{g}(\mathcal{T}_B)\right]. \tag{17}
$$

Denote the Jacobian matrix of the logits $f(\boldsymbol{x}; \boldsymbol{\theta})$ w.r.t. $\boldsymbol{\theta}$ by $\boldsymbol{J}_{\boldsymbol{\theta}}(\boldsymbol{x})$, i.e.,

$$
\boldsymbol{J}_{\boldsymbol{\theta}}(\boldsymbol{x}) = \left[\nabla_{\boldsymbol{\theta}} f_1(\boldsymbol{x}; \boldsymbol{\theta}), \nabla_{\boldsymbol{\theta}} f_2(\boldsymbol{x}; \boldsymbol{\theta}), ..., \nabla_{\boldsymbol{\theta}} f_c(\boldsymbol{x}; \boldsymbol{\theta})\right], \tag{18}
$$

where $f_j(\boldsymbol{x}; \boldsymbol{\theta})$ is the $j$-th entry of $f(\boldsymbol{x}; \boldsymbol{\theta})$, and $c$ is the number of classes.

Using first order approximation, we have:

$$
f(\boldsymbol{x}; \boldsymbol{\theta} + g(\mathcal{T}_B)) \approx f(\boldsymbol{x}; \boldsymbol{\theta}) + \boldsymbol{J}_{\boldsymbol{\theta}}(\boldsymbol{x})^T g(\mathcal{T}_B), \tag{19}
$$

and $OV(\boldsymbol{x})$ can be written as:

$$
OV(\boldsymbol{x}) \approx \frac{\mathbb{E}_{\mathcal{T}_B}\left[\tilde{g}(\mathcal{T}_B)^T \boldsymbol{J}_{\boldsymbol{\theta}}(\boldsymbol{x}) \boldsymbol{J}_{\boldsymbol{\theta}}(\boldsymbol{x})^T \tilde{g}(\mathcal{T}_B)\right]}{f(\boldsymbol{x}; \boldsymbol{\theta})^T f(\boldsymbol{x}; \boldsymbol{\theta}) + \mathbb{E}_{\mathcal{T}_B}\left[O\left(\|g(\mathcal{T}_B)\|_2\right)\right]} \approx \frac{\mathbb{E}_{\mathcal{T}_B}\left[\tilde{g}(\mathcal{T}_B)^T \boldsymbol{J}_{\boldsymbol{\theta}}(\boldsymbol{x}) \boldsymbol{J}_{\boldsymbol{\theta}}(\boldsymbol{x})^T \tilde{g}(\mathcal{T}_B)\right]}{f(\boldsymbol{x}; \boldsymbol{\theta})^T f(\boldsymbol{x}; \boldsymbol{\theta})}. \tag{20}
$$

The only difference between $\mathbb{E}_{\boldsymbol{x}}\left[OV(x)\right]$ and $V_g$ is the middle weight matrix $\mathbb{E}_{\boldsymbol{x}}\left[\frac{\boldsymbol{J}_{\boldsymbol{\theta}}(\boldsymbol{x}) \boldsymbol{J}_{\boldsymbol{\theta}}(\boldsymbol{x})^T}{f(\boldsymbol{x}; \boldsymbol{\theta})^T f(\boldsymbol{x}; \boldsymbol{\theta})}\right]$. This suggests that penalizing the gradient variance can also reduce the optimization variance.

Figure 8 presents the curves of $V_g$ in the training procedure. It can be observed that $V_g$ also shows some ability to indicate the generalization performance. However, compared with the results in Figure 2, we can see that OV demonstrates a stronger power for indicating the generalization error than $V_g$. More importantly, $V_g$ loses its comparability when the network size increases, while OV can be more reliable to architectural changes with the middle weight matrix $\mathbb{E}_{\boldsymbol{x}} \left[ \frac{\boldsymbol{J_\theta}(\boldsymbol{x})\boldsymbol{J_\theta}(\boldsymbol{x})^T}{f(\boldsymbol{x};\boldsymbol{\theta})^T f(\boldsymbol{x};\boldsymbol{\theta})} \right]$ to normalize $V_g$, which is illustrated in Figure 7.

We also notice that $\|\mathbb{E}_{\mathcal{T}_B} g(\mathcal{T}_B)\|_2^2$ is usually far less than $\mathbb{E}_{\mathcal{T}_B} \|g(\mathcal{T}_B)\|_2^2$, hence $V_g$ and the gradient norm $\mathbb{E}_{\mathcal{T}_B} \|g(\mathcal{T}_B)\|_2^2$ almost present the same curves in the training procedure.

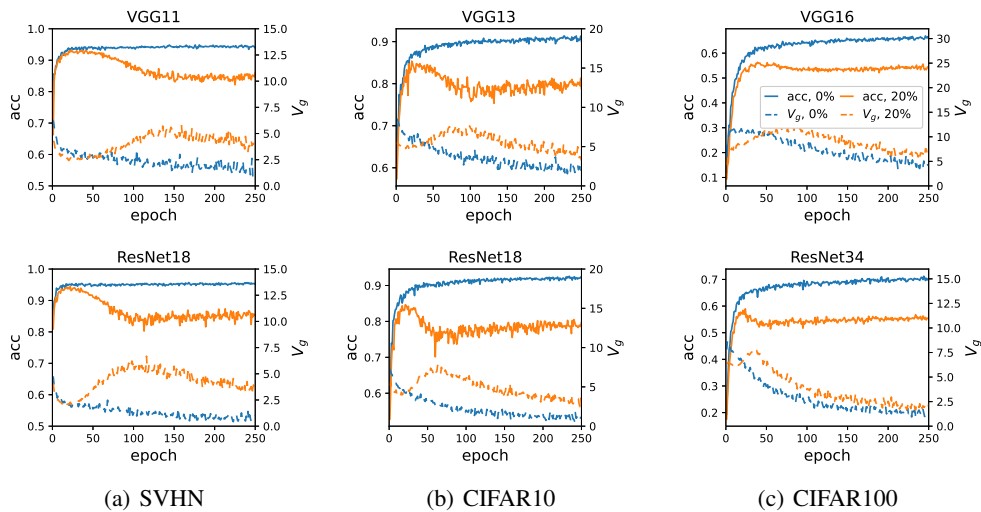

Figure 8: Test accuracy and $V_g$. The models were trained with the Adam optimizer (learning rate 0.0001). The number in each legend indicates its percentage of label noise.

## C  BEHAVIORS OF DIFFERENT LOSS FUNCTIONS

Many different loss functions can be used to evaluate the test performance of a model. They may have very different behaviors w.r.t. the training epochs. As shown in Figure 9, the epoch-wise double descent can be very conspicuous on test error, i.e., the ZO loss, but barely observable on CE and MSE losses, which increase after the early stopping point. This is because at the late stage of training, model outputs approach 0 or 1, resulting in the increase of the CE and MSE losses on the misclassified test samples, though the decision boundary may be barely changed. When rescaling the weights of the last layer by a positive real number, the ZO loss remains the same because of the untouched decision boundary, whereas the CE and MSE losses are changed. Thus, we perform bias-variance decomposition on the ZO loss to study epoch-wise double descent.

## D  VGG11 ON SVHN BY ADAM OPTIMIZER WITH LEARNING RATE 0.001

Training VGG11 on SVHN by Adam optimizer with learning rate 0.001 is unstable, as shown in Figure 14(a). Figure 10 shows the test error and optimization variance. For 0% and 10% label noise, the test error stays large (the test accuracy is low) for a long period in the early phase of training. The optimization variance is also abnormal.

## E  LOSS, BIAS AND VARIANCE W.R.T. DIFFERENT LEVELS OF LABEL NOISE

Label noise makes epoch-wise double descent more conspicuous to observe (Nakkiran et al., 2020). If the variance is the major cause of double descent, it should match the variation of the test error

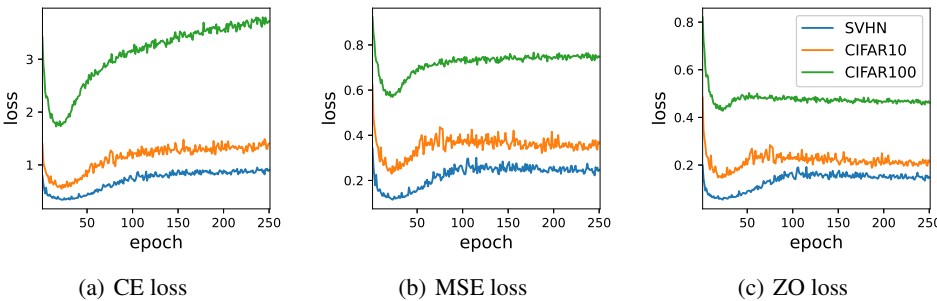

Figure 9: Different loss functions w.r.t. the training epoch. ResNet18 was trained on SVHN, CIFAR10, and CIFAR100 with 20% label noise to introduce epoch-wise double descent. Adam optimizer with learning rate 0.0001 was used.

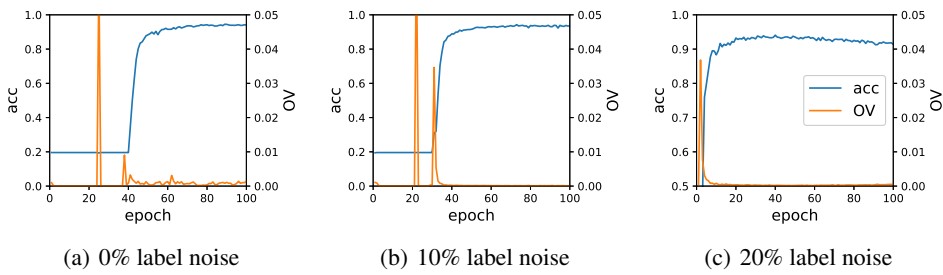

Figure 10: Test accuracy and optimization variance (OV) of VGG11 on SVHN, w.r.t. different levels of label noise. **Adam** optimizer with learning rate 0.001 was used.

when adding different levels of label noise. Figure 11 shows an example to compare the loss, variance, and bias w.r.t. different levels of label noise. Though label noise impacts both the bias and the variance, the latter appears to be more sensitive and shows better synchronization with the loss. For instance, when we randomly shuffle a small percentage of labels, say 10%, a valley clearly occurs between 20 and 50 epoches for the variance, whereas it is less obvious for the bias. In addition, it seems that the level of label noise does not affect the epoch at which the loss reaches its first minimum. This is surprising, because the label noise is considered highly related to the complexity of the dataset. Our future work will explore the role label noise plays in the generalization of DNNs.

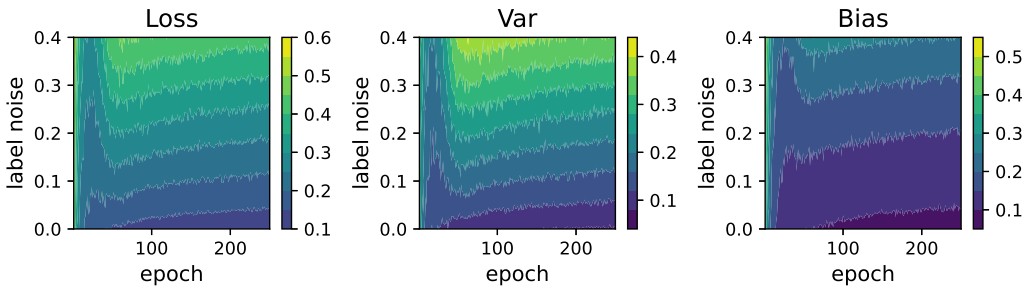

Figure 11: Loss, variance and bias w.r.t. different levels of label noise. The model was ResNet18 trained on CIFAR10. Adam optimizer with learning rate 0.0001 was used.

# F  BIAS AND VARIANCE TERMS W.R.T. DIFFERENT LOSS FUNCTIONS

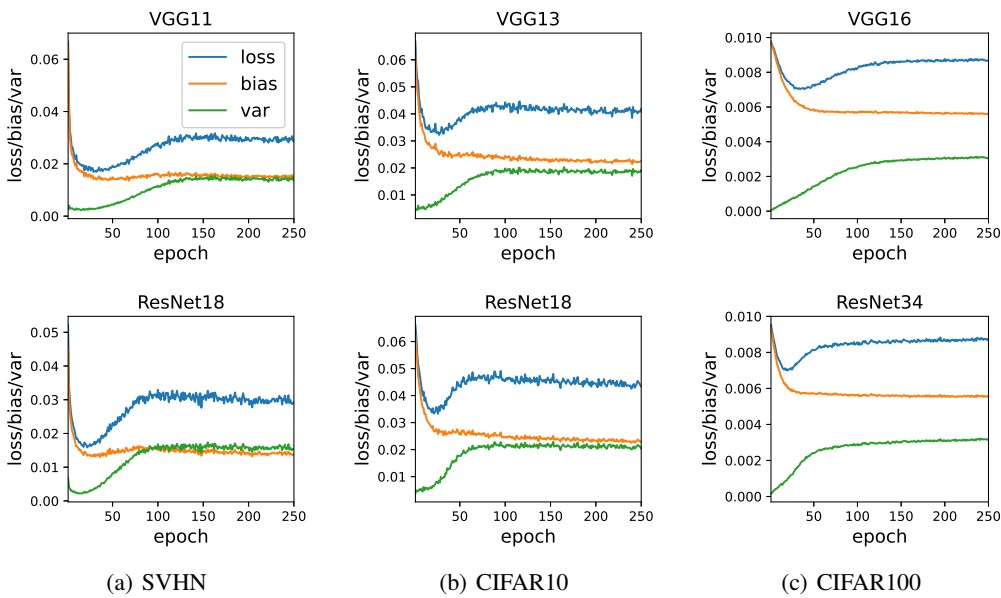

Figure 12: Test **MSE** loss and the corresponding bias/variance terms. The models were trained with 20% label noise. Adam optimizer with learning rate 0.0001 was used.

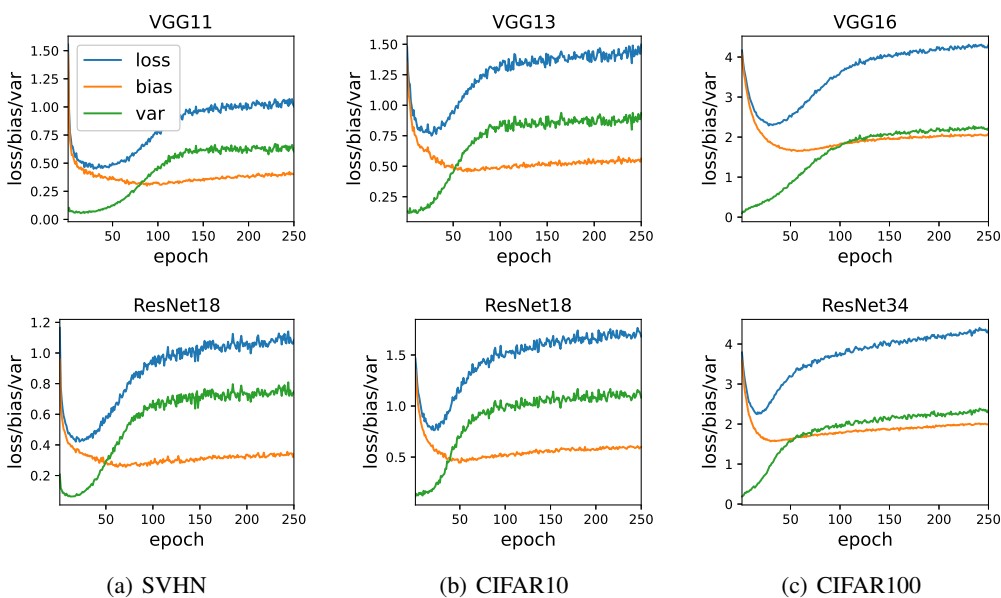

Figure 13: Test **CE** loss and the corresponding bias/variance terms. The models were trained with 20% label noise. Adam optimizer with learning rate 0.0001 was used.

## G  BIAS AND VARIANCE TERMS W.R.T. DIFFERENT OPTIMIZERS AND LEARNING RATES

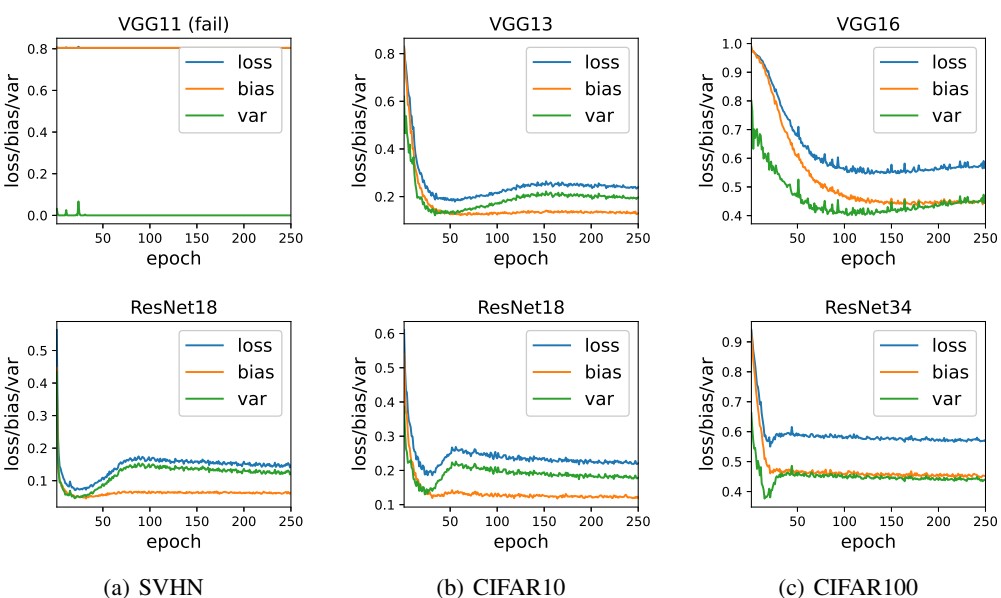

Figure 14: The expected test ZO loss and its bias and variance. The models were trained with 20% label noise. **Adam** optimizer with learning rate **0.001** was used.

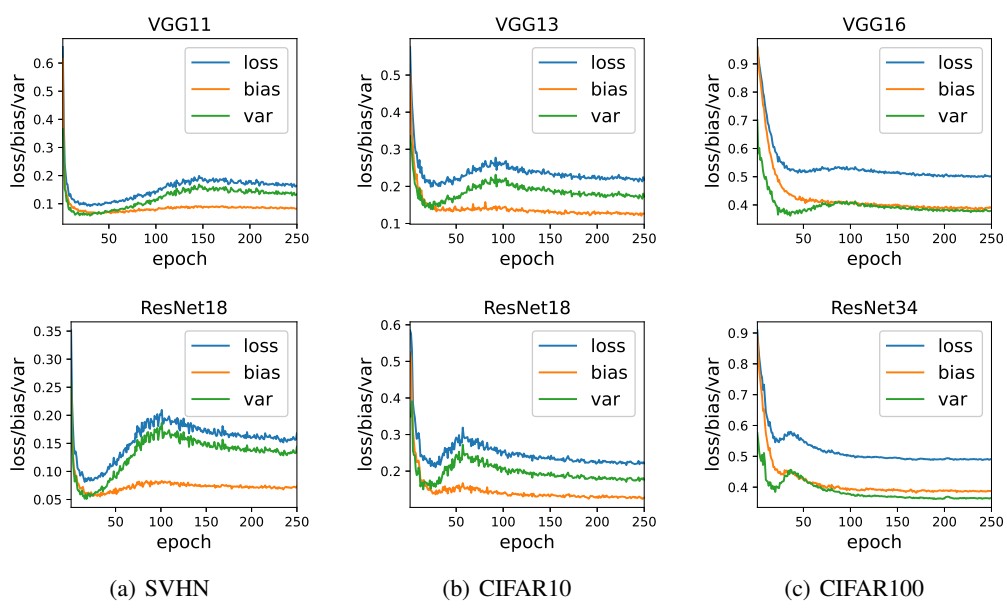

Figure 15: Expected test ZO loss and its bias and variance. The models were trained with 20% label noise. **SGD** optimizer (momentum $= 0.9$) with learning rate **0.01** was used.

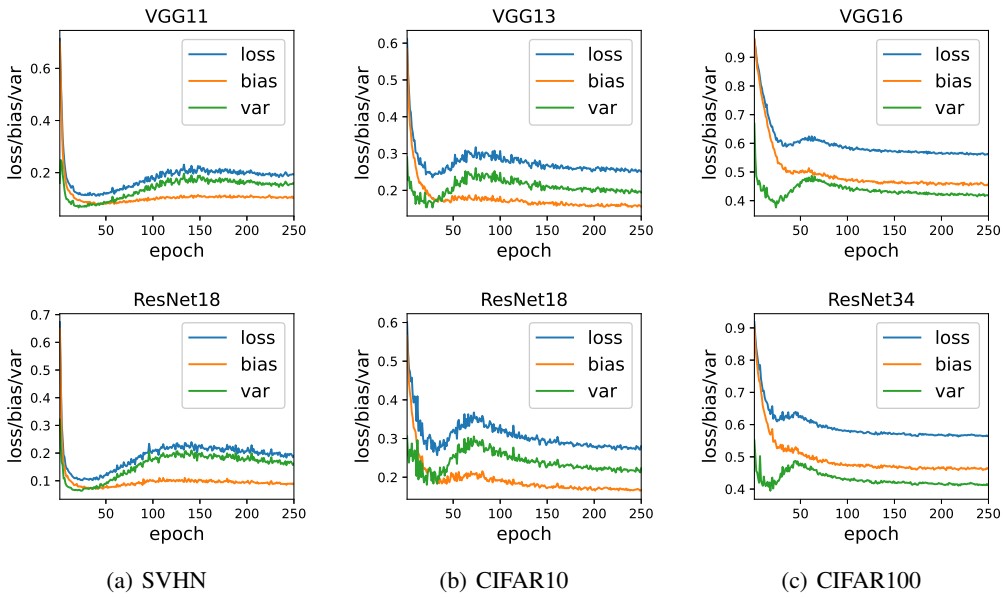

Figure 16: Expected test ZO loss and its bias and variance. The models were trained with 20% label noise. **SGD** optimizer (momentum $= 0.9$) with learning rate **0.001** was used.

# H    DETAILED INFORMATION OF THE SMALL CNN MODEL

We present the detailed information of the architecture trained with small numbers of training samples. It consists of two convolutional layers and two fully-connected layers, as shown in Table 2.

Table 2: Architecture of the small CNN model ("BN" is short for Batch Normalization).

| Layers | Parameters | BN | Activation | Max pooling |
|--------|-----------|----|-----------|-------------|
| Input | input size=(32, 32)×3 | - | - | - |
| Conv | filters=(3, 3)×32; | ✓ | ReLU | (2, 2) |
| Conv | filters=(3, 3)×64; | ✓ | ReLU | (2, 2) |
| Dense | nodes=1024 | - | ReLU | - |
| Dense | nodes=10 | - | Softmax | - |

# I  OPTIMIZATION VARIANCE AND TEST ACCURACY W.R.T. DIFFERENT OPTIMIZERS AND LEARNING RATES

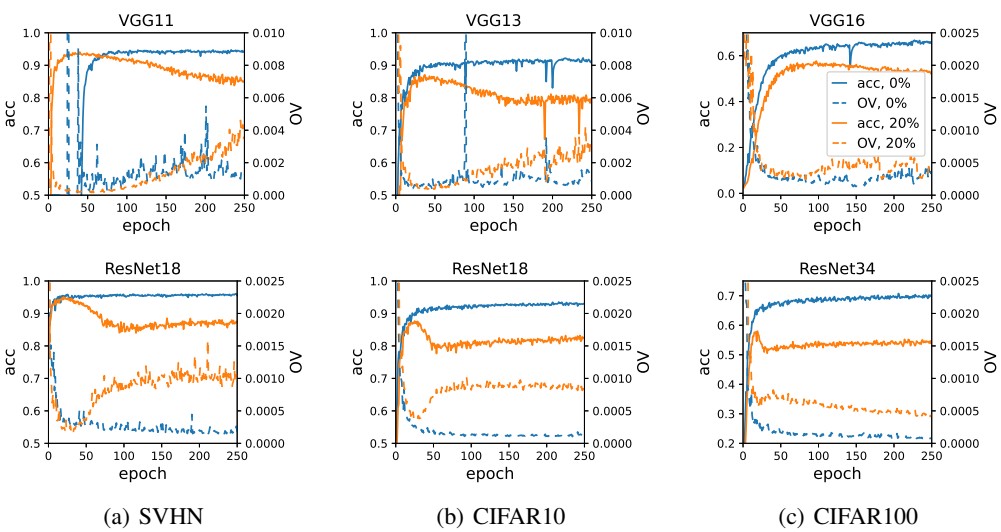

Figure 17: Test accuracy and optimization variance (OV). The models were trained with **Adam** optimizer (learning rate 0.001). The number in each legend indicates its percentage of label noise.

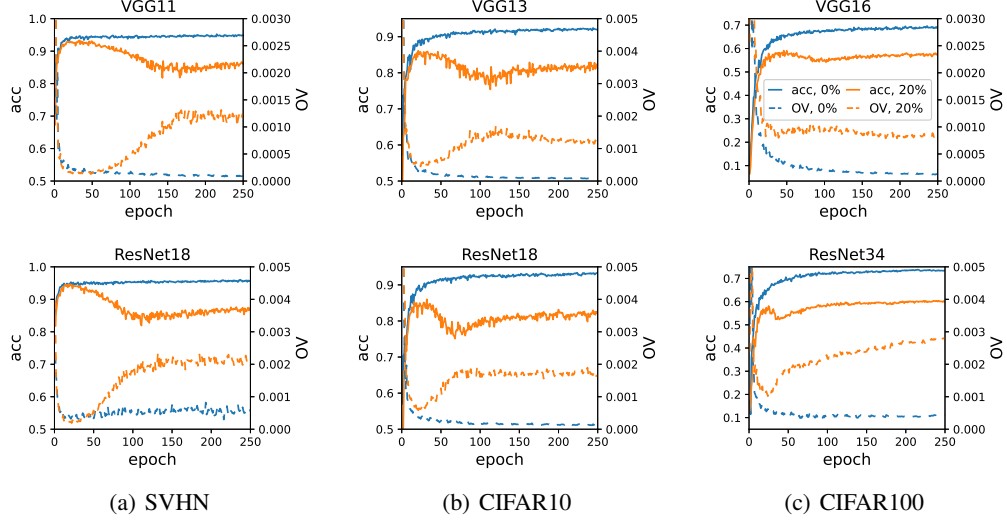

Figure 18: Test accuracy and optimization variance (OV). The models were trained with **SGD** optimizer (learning rate 0.01, momentum 0.9). The number in each legend indicates its percentage of label noise.

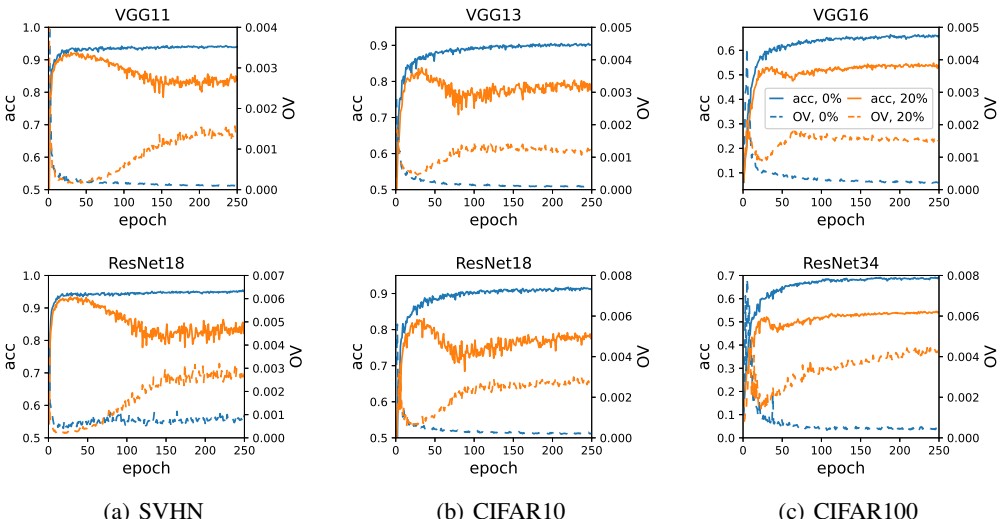

(a) SVHN           (b) CIFAR10           (c) CIFAR100

Figure 19: Test accuracy and optimization variance (OV). The models were trained with the **SGD** optimizer (learning rate 0.001, momentum 0.9). The number in each legend indicates its percentage of label noise.

