# OpenReview forum: "Optimization Variance:  Exploring Generalization Properties of DNNs"
_ICLR.cc/2021/Conference — Reject_

### Official Review · AnonReviewer2 · 2020-10-26

**Rating:** 5
**Confidence:** 4

**Review:**

The paper under review studies the epoch wise double descent phenomena empirically. The epoch wise double descent phenomena is the observation that the risk of a large neural network trained with SGD first decreases, then increases, and finally decreases again as a function of the epochs or SGD steps. In addition, it proposes a quantity called ``optimization variance (OV)'', and it demonstrate that OV correlates with the test error. Based on this observation, it proposes to early stop when the OV reaches a minimum.

Strong points are that the paper
i) is an early work studying epoch-wise double descent, and that
ii) the newly introduced optimization variance is an interesting concept and might very well be useful for finding good early stopping points.

Weak points are that
i) the findings about epoch-wise double descent are not conclusive, and that
ii) optimization variance is not sufficiently evaluated to judge its usefulness. Both is discussed in more detail in Comments 1-6 below.

Based on comments 1-6 below I think that the paper's conclusions are not yet supported by the papers experiment. I do think that the paper can be developed into a very nice contribution, if more substantive findings about epoch-wise double descent can be obtained, and/or if the value of the newly introduced optimization variance is evaluated more thoroughly (e.g., through simple theory, or through more rigorous experiments).

Comments:
1/ The paper's main finding on epoch-wise double descent is that ``the variance dominates the epoch-wise double descent of the test error'' meaning that it ``is mainly the variance that contributes to epoch-wise double descent''. This statement is overly general, because it is drawn from observations for one particular setup (20\% label noise, CIFAR10 and SVHN). If the label noise is less, it becomes apparent that both the variance and bias work together (as a sum) to generate a double-descent like curve. Adding curves for other distributions (e.g., less label noise, other datasets) can illustrate that.

2/ The paper's main contribution to the study of epoch-wise double descent is the empirical bias-variance decomposition. However, this approach does not enable to draw conclusions on why double descent occurs, as the paper notes itself ``the reason why the variance begins to diminish at the late phase of training is still unclear''.

3/ The paper finds a strong empirical correlation between the test error and the newly introduced optimization variance (OV) (Fig. 2), and proposes to early stop based on the OV. However, Fig. 2 also shows when this fails: c) ResNet 34 on CIFAR-10 shows that OV is minimal at 250 epochs, while the accuracy is highest after about 15-20 epochs. Of course, such an early stopping rule is allowed to fail sometimes, but the evidence that this is a good early stopping rule is not convincing.

4/ The value of OV would be much more convincing if the paper could provide a theoretical result even for a simple toy case like a linear model.
It would also be more convincing if the benefits of OV hold for a broad set of models; the paper points out that it doesn't generalize to a large class of models: ``we need to point out that we did not observe a strong correlation between the OV and the test accuracy when using significantly different architectures''.

5/ The paper omits a recent reference on epoch-wise double descent: https://arxiv.org/abs/2007.10099, which is very related because it explains epoch-wise double descent as overlapping bias variance tradeoffs for a variety of setups, both empirically and practically.

6/ As https://arxiv.org/abs/2007.10099 finds, epoch-wise double descent can bit mitigated, and this can typically improves performance. How does the paper's approach to find the early stopping point work in this setup, where double descent is mitigated?

Minor comments:
``20\% labels of the training data were randomly shuffled to strengthen epoch-wise double descent'': This is misleading as it suggested that with less or zero label noise, epoch-wise double descent does occur, but it does not for the models and setup considered in the paper.

-----
UPDATE: Thanks for the response, I have responded below and kept the score constant.

---

> ### Author Response · Authors · 2020-11-21
> **Response to Reviewer 2 - Part 1**
>
> Thanks for your helpful comments! Our detailed replies are given below.
>
> - *The paper's main finding on epoch-wise double descent is that the variance dominates the epoch-wise double descent of the test error'' meaning that it is mainly the variance that contributes to epoch-wise double descent''. This statement is overly general, because it is drawn from observations for one particular setup (20% label noise, CIFAR10 and SVHN). If the label noise is less, it becomes apparent that both the variance and bias work together (as a sum) to generate a double-descent-like curve. Adding curves for other distributions (e.g., less label noise, other datasets) can illustrate that.*
>
>   Thanks for your suggestion. However, we need to emphasize that our conclusion is not just drawn from a single experiment setup but from *a series of experiments with different levels of label noises*, as shown in Appendix E of the original version.
>
>   Indeed the double-descent-like curve is caused by both the bias and variance, but as shown in Figure 1 in the main content and Figure 9 in the Appendix E, variance plays a significantly more dominant role in the case of the zero-one loss. This is also one of our novel findings that contradicts previous findings on CE and MSE losses, where only combining bias and variance can lead to the double-descent-like curves (Appendix F).
>
>   Another point our paper would like to demonstrate is that the zero-one loss is a better choice for analyzing the epoch-wise double descent, compared to MSE and CE losses. The major reason is that both MSE and CE losses can be easily manipulated without changing the decision boundary. In contrast, the zero-one loss is more directly related to the generalization performance, as shown in Appendix C.
>
> - *The paper's main contribution to the study of epoch-wise double descent is the empirical bias-variance decomposition. However, this approach does not enable to draw conclusions on why double descent occurs, as the paper notes itself ``the reason why the variance begins to diminish at the late phase of training is still unclear'.*
>
>   We would like to clarify that our focus of this paper is not understanding why double-descent occurs. We include the discussion of epoch-wise double descent to motivate the potential of establishing connections between optimization stability of a DNN and its generalization errors, where we found the variance is a good indicator of generalization errors even for a complex error curve like epoch-wise double descent. As shown in our experiments, the proposed OV well indicates the change of generalization errors for cases having epoch-wise double descent. Moreover, epoch-wise double descent is not a necessary condition for applying OV. As shown in Figure 2 (blue curves), we compared the values of OV and generalization errors of DNNs when there are 0% label noises, from which we can see the curves of generalization errors have no epoch-wise double descent, yet the proposed OV still works pretty well. We will add this discussion to the revision.

---

> > ### Comment · AnonReviewer2 · 2020-11-22
> > **comment on response**
> >
> > Thanks for revising the manuscript, and for the response, in particular for pointing out that the paper provides results in appendix E for different noise setups, which I missed when reading the manuscript. I also agree that the variance seems to play a more dominant role, but ultimately the bias and variance play together to generate the risk curve, even for zero-one loss.

---

> ### Author Response · Authors · 2020-11-21
> **Response to Reviewer 2 - Part 2**
>
> - *The paper finds a strong empirical correlation between the test error and the newly introduced optimization variance (OV) (Fig. 2), and proposes to early stop based on the OV. However, Fig. 2 also shows when this fails: c) ResNet 34 on CIFAR-10 shows that OV is minimal at 250 epochs, while the accuracy is highest after about 15-20 epochs. Of course, such an early stopping rule is allowed to fail sometimes, but the evidence that this is a good early stopping rule is not convincing. The value of OV would be much more convincing if the paper could provide a theoretical result even for a simple toy case like a linear model. It would also be more convincing if the benefits of OV hold for a broad set of models; the paper points out that it doesn't generalize to a large class of models: ``we need to point out that we did not observe a strong correlation between the OV and the test accuracy when using significantly different architectures''.*
>
>   We are sorry for the confusion. By saying “we need to point out that we did not observe a strong correlation between the OV and the test accuracy when using significantly different architectures”, we did not mean OV cannot generalize to a large class of models; instead, it means that we cannot compare the test accuracy of significantly different architectures by comparing their OV values directly. In fact, the effectiveness of OV can be generally observed. We will revise our paper to make this claim clearer.
>
>   Although our paper lacks a rigorous theoretical justification to illustrate the connection between OV and generalization, in this paper we provide a comprehensive study to empirically demonstrate the effectiveness of OV. As shown in our experiments, OV can be a good indicator of the generalization errors without the use of validation set, which we believe may shed the light for many valuable research directions and applications. The theoretical analysis of OV is left as an important future work and we plan to study the correlation between the optimization stability and generalization performance theoretically in the PAC-Bayesian framework.
>
> - *The paper omits a recent reference on epoch-wise double descent: https://arxiv.org/abs/2007.10099, which is very related because it explains epoch-wise double descent as overlapping bias variance tradeoffs for a variety of setups, both empirically and practically.*
>
>   This is a very interesting study, highly relevant to our topic. Thanks for pointing this out. We will discuss this paper in our revision.
>
> - *As https://arxiv.org/abs/2007.10099 finds, epoch-wise double descent can bit mitigated, and this can typically improves performance. How does the paper's approach to find the early stopping point work in this setup, where double descent is mitigated?*
>
>   Thanks for your suggestion! We are now working on checking the OV under this experimental setup. We will update our paper to include this experiment once it’s done.
>
> - *Minor comments: ``20% labels of the training data were randomly shuffled to strengthen epoch-wise double descent'': This is misleading as it suggested that with less or zero label noise, epoch-wise double descent does occur, but it does not for the models and setup considered in the paper.*
>
>   Thanks for pointing this out. We will revise this sentence to make it more accurate.

---

### Official Review · AnonReviewer4 · 2020-10-28
**interesting idea, but needs more work**

**Rating:** 5
**Confidence:** 3

**Review:**

* quality
The idea is interesting, but the paper lacks theoretical insights. And it was difficult to find in the paper how OV is related to generalization.
* clarity
The paper needs to make it more clear how OV is related to generalization - is it always monotonically decreasing/increasing?
* originality
The idea seems new, but more theoretical insight is needed.
* significance
If the authors include more theoretical justification, the paper's results would be more significant. The current version is too empirical and not very convincing.

---

> ### Author Response · Authors · 2020-11-21
> **Response to Reviewer 4**
>
> Thanks for your insightful comments. Please see our detailed responses below.
>
> - *The idea is interesting, but the paper lacks theoretical insights. And it was difficult to find in the paper how OV is related to generalization. The paper needs to make it more clear how OV is related to generalization - is it always monotonically decreasing/increasing?*
>
>   Thanks for your suggestion, although our paper lacks a rigorous theoretical justification to illustrate the connection between OV and generalization, in this paper we provide a comprehensive study to empirically demonstrate the effectiveness of OV. As shown in our experiments, OV can be a good indicator of the generalization errors without the use of validation set, which we believe may shed the light for many valuable research directions and applications.
>
>   The intuition of OV is quite clear. OV can be regarded as a metric that measures the function robustness of DNNs with respect to sampling noises, and sampling noises are the major cause of optimization variance -- how drastically the function captured by a DNN varies as the optimization proceeds. Consequently, when the value of OV is large, the generalization error of the DNN is very likely to be bad, since the function learned by DNN has a large variance.
>
>   A similar metric of OV is the sharpness of local minima proposed in [1], which measures the stability of local minima as an indicator of the generalization error. OV can be treated as an extension of this metric, which measures the stability of a DNN during the entire optimization process, instead of just for local minima.
>
>   Our understanding of the comment “is it always monotonically decreasing/increasing?” is concerning how the value of OV will change. As shown in our experiments (Figure 2, Figure 4, Figure 5, etc), the value of OV will neither monotonically decrease nor increase. Instead, it changes consistently with the generalization errors, which has been verified across multiple datasets and network architectures.
>
>   [1] Keskar et al. On large-batch training for deep learning: Generalization gap and sharp minima. ICLR 2017.
>
> - *The current version is too empirical and not very convincing.*
>
>   Thanks for your constructive opinion! While theoretical analysis is left as an important future work where we plan to study in the PAC-Bayesian framework.
>
>   To make our findings convincing, we performed several experiments on different architectures with different optimizers to verify the effectiveness of OV. We also show that OV can indicate the generalization performance when increasing the network size. Additionally, as suggested by Reviewer 3, we conducted experiments to examine the variation of OV with different sizes of the training set, whose results show that OV also correlates well with the generalization errors as a function of the size of training sets. All these results verified the effectiveness of our proposed metric.

---

### Official Review · AnonReviewer3 · 2020-10-29
**Validation datasets are no longer needed?**

**Rating:** 7
**Confidence:** 5

**Review:**

Having a stopping rule without the validation set is intriguing, especially for datasets with a low number of samples. The authors propose a rule that doesn't require the validation dataset, i.e. it is solely based on training data. It introduces the notion of optimization variance which is different from the variance of gradients.
I give them credit for the idea and also the analytical comparison with the variance of gradients.

On the one hand, the experiments in Section 3.3 are geared towards showing the role of the validation dataset, but on the other hand they lack rigor. It would be interesting to learn the impact at different levels of the percentage of the samples in validation. I also think the work can be impactful on small datasets. While they experiment with large-scale datasets, experiments with even smaller datasets and different validation set proportions would be of great interest.

---

> ### Author Response · Authors · 2020-11-21
> **Response to Reviewer 3**
>
> Thanks for your helpful and positive comments! Our detailed replies are given below.
>
> - *On the one hand, the experiments in Section 3.3 are geared towards showing the role of the validation dataset, but on the other hand they lack rigor. It would be interesting to learn the impact at different levels of the percentage of the samples in validation. I also think the work can be impactful on small datasets. While they experiment with large-scale datasets, experiments with even smaller datasets and different validation set proportions would be of great interest.*
>
>   The comment is very helpful. As suggested, we performed experiments using smaller numbers of training samples (2000, 4000, 6000) on CIFAR10, and the results are included in Section 3.4. Our results show two important phenomena:
> a) When the size of the training set is small, OV still correlates well with the generalization performance as a function of the training epochs, which verifies the validity of our results on small datasets.
> b) As expected, more training samples usually lead to better generalization performance, which can also be reflected by comparing the values of OV.

---

### Official Review · AnonReviewer1 · 2020-10-31
**Hard to see the novelty of the metric and how is OV different from simpler metric**

**Rating:** 5
**Confidence:** 3

**Review:**

This paper proposes the metric of estimating the variance of gradients during optimization of deep networks and empirically found that this metric correlates with test errors, which may indicate a good point for performing early stopping during training.

My first thinking is that the metric is indeed interesting and informative if, it could be applied to indicates the generalization on the test set, of course. Then my concern rises from the methodology used in this paper, as using 'a correlation' between optimization variance and generalization error to indicate a 'causal relation' between these two numbers. Granted, it might be difficult to actually prove it. From a big picture, it would still be nice to see a comparison analysis with other simpler metrics, such as the gradient magnitude itself?

The definition of "optimization variance" seems to be weird, as in equation 6, the nominator scales with gradient magnitudes, and the denominator tends to have very smaller changes after some iterations, comparably. So why is this OV so different from the gradient norms as to constitute a novel metric? At least from experiments, the changes in denominator do not matter that much.

The dataset is kind of simple in the empirical analysis. It would be better if Imagenet size level ones or regression tasks are also performed.

Small questions:

Figure 3, what is 'different number of training batches'? is that batch size?

Figure 4 shows some strong upheaval in terms of accuracy, is that from a large gradient that somehow explodes during training?

Figure 5 has really unreadable captions and legend notations.

---

> ### Author Response · Authors · 2020-11-21
> **Response to Reviewer 1 - Part 1**
>
> Thanks for your helpful comments! Our detailed replies are given below.
>
> - *This paper proposes the metric of estimating the variance of gradients during optimization of deep networks and empirically found that this metric correlates with test errors, which may indicate a good point for performing early stopping during training.
>   My first thinking is that the metric is indeed interesting and informative if, it could be applied to indicates the generalization on the test set, of course. Then my concern rises from the methodology used in this paper, as using 'a correlation' between optimization variance and generalization error to indicate a 'causal relation' between these two numbers. Granted, it might be difficult to actually prove it. From a big picture, it would still be nice to see a comparison analysis with other simpler metrics, such as the gradient magnitude itself?*
>
>   Indeed, it is hard to rigorously indicate the causal relationship between OV and the generalization performance. However, the insight behind OV may give some clue. Specifically,  OV can be regarded as a metric that measures the function robustness of DNNs to sampling noises, and sampling noises are the major cause of optimization variance -- how drastically the function captured by a DNN varies as the optimization proceeds. Consequently, when the value of OV is large, the generalization error of the DNN is very likely to be bad, since the function learned by DNN has a large variance.
>
>   A similar metric of OV is the sharpness of local minima proposed in [1], which measures the robustness of local minima as an indicator of the generalization error. OV can be treated as an extension of this metric, which measures the stability of a DNN during *the entire optimization process*, instead of just for local minima.
>
>   As suggested, a comparison between OV and gradient norm is included in the revision (Appendix B) . From the results we can see OV demonstrates a stronger power for indicating the generalization error than gradient norm. More importantly, the gradient norm loses its comparability when the network size increases, while OV can be more reliable to architectural changes as shown in Section 3.5. We will add these discussions in the revision.
>
>   [1] Keskar et al. On large-batch training for deep learning: Generalization gap and sharp minima. ICLR 2017.
>
> - *The definition of "optimization variance" seems to be weird, as in equation 6, the nominator scales with gradient magnitudes, and the denominator tends to have very smaller changes after some iterations, comparably. So why is this OV so different from the gradient norms as to constitute a novel metric? At least from experiments, the changes in denominator do not matter that much.*
>
>   The intuition of OV actually comes from the definition of  *coefficient of variation* (CV) in probability theory and statistics, which is also known as the *relative standard deviation* (https://en.wikipedia.org/wiki/Coefficient_of_variation). CV is defined as the ratio between the standard deviation $\sigma$ and the mean $\mu$, and is independent of the unit in which the measurement is taken. Hence CV enables comparing the relative diversity between two different measurements.
>
>   In terms of OV, the variance of logits, i.e., the numerator of OV, is not comparable across epochs due to the influence of their norm. In fact, even if the variance of logits maintains the same during the whole optimization process, its influence on the decision boundary is limited when the logits are large. Consequently, by treating the norm of logits as the measurement unit, following CV we set OV to $\frac{\sum_i\sigma_i^2}{\sum_i\mu_i^2}$ ($i$ represents the entry of logits).  If we remove the denominator, the value of OV will no longer have the indication ability for generalization error, especially at the early stage of the optimization process.
>
>   Thanks for pointing this out, and we will include this insight behind OV’s formulation in the revision.
>
> - *The dataset is kind of simple in the empirical analysis. It would be better if Imagenet size level ones or regression tasks are also performed.*
>
>   We follow the standard practice to include SVHN, CIFAR10 and CIFAR100 in our evaluation to analyze the generalization performance of DNNs. For a large dataset like ImageNet, a validation set can be easily partitioned from the training set without hurting the generalization performance. Therefore, OV may be more useful in small datasets. As suggested by Reviewer 3, we further performed experiments with even fewer training samples (2000, 4000, 6000) on CIFAR10, and found that OV also works pretty well (See Section 3.4 in the revision).
>
>   Thanks for your suggestion on evaluating OV on regression tasks. As pointed out in the discussion, evaluating OV on regression tasks is left as the future work.

---

> ### Author Response · Authors · 2020-11-21
> **Response to Reviewer 1 - Part 2**
>
> - *Small questions:*
>
>   *1. Figure 3, what is 'different number of training batches'? is that batch size?*
>
>   It means the number of training batches used to calculate the gradients when estimating OV, instead of the batch size. It has been mentioned in Page 6
>
>   *2. Figure 4 shows some strong upheaval in terms of accuracy, is that from a large gradient that somehow explodes during training?*
>
>   Upheaval may look stronger than what it actually is, because the range of the Y-axis is [0.97, 0.99]. Nevertheless, it does come from a larger OVcaused by a larger gradient, as you pointed out.
>
>   *3. Figure 5 has really unreadable captions and legend notations.*
>
>   Thanks for pointing this out! We have detailed the caption and legend notations in the revision.

---

### Official Review · AnonReviewer5 · 2020-11-08
**The paper is in the right direction, but more work needed to be more covnvincing.**

**Rating:** 5
**Confidence:** 4

**Review:**

Summary:

The paper studies the trajectory of the test error as a function of training time focusing on Epoch-Wise Double-Descent.  Similar to "Rethinking Bias-Variance Trade-off for Generalization of Neural Networks" by Yang et. al., the paper shows that if one decomposes the test error to bias and variance terms, Double Descent occurs as a function of train time as a result of unimodality of the variance term (while the bias term decreases monotonically).  The paper also introduces a quantity they name optimization variance (OV) and that correlates with the test error (while being only a function of the train set) and can be useful for early stopping.

=====================================================

Main Concerns and improvement points:
Regarding the variance as the main culprit behind Double Descent:
While it is interesting to see that the analysis of Yang et. al. carries over to epoch double descent, by itself, this experiment is not too surprising. Citing the discussion section "Contradicting to the traditional view that the variance keeps increasing because of overfitting, our experimental results show a more complex behavior: the variance starts high and then decreases
rapid". This is exactly the claim made in Yang et. al, for model double descent making the novelty of the empirical result more limited. As the authors mention (and promise to study in the future), understanding why the variance diminished with time after the interpolation threshold is an important question that could substantially strengthen the paper.

Regarding Optimization Variance:
While it is cool that the optimization variance not too hard to compute and correlates well with the test set (for a fixed architecture) I'm not sure what is the fundamental contribution here. Just defining a quantity and showing some properties it has without motivation for why and when should we expect this quantity to arise is not a persuasive result. A couple of specific concerns for me are:
1) Correlation does not mean much by itself. For example, the train accuracy is very correlated with the test accuracy. There are other empirical quantities that correlate with the test (and provably so at times, see for example https://arxiv.org/pdf/1912.00528.pdf).
2) Usually, training for long enough is sufficient to have high correlation with the test.  For example in fig 5 we could ask where would lie a model trained for a very long time (say 100 epochs) my bet it would right there on the line of best test error.
3) If the argument here is early stopping I can just use the test set.
4) The quantity does not correlate through different model architectures.

To alleviate my concerns I would love to see some theoretical justification for *why* is this an important quantity and when does it arise naturally. Alternatively, if we can predict test set across different model architectures, that would give more motivation to as why we might want to look at this quantity more deeply.

To summarize, the first part of the paper is lacking in novelty and in the second part, the OV is not deeply enough motivated.
I do believe that the paper is on the right track and with some work should be publishable in a future conference.

Minor Comments:
It might be a personal preference, but I find the notation confusing and in disconnect with the standard notation in the field. For example, using t as labels and y as predictions is highly non-standard and would confuse many readers.  (why not y for label, \hat y or p for prediction, t for training time?)
"The expected loss should be small to ensure good generalization performance" - this is a very informal sentence that I found confusing rather than helpful.

---

> ### Author Response · Authors · 2020-11-18
> **Response to Reviewer 5 - Part 1**
>
> Thank you for your constructive feedback! Please see our detailed response below.
>
> - *Main Concerns and improvement points: Regarding the variance as the main culprit behind Double Descent: While it is interesting to see that the analysis of Yang et. al. carries over to epoch double descent, by itself, this experiment is not too surprising. Citing the discussion section "Contradicting to the traditional view that the variance keeps increasing because of overfitting, our experimental results show a more complex behavior: the variance starts high and then decreases rapid". This is exactly the claim made in Yang et. al, for model double descent making the novelty of the empirical result more limited. As the authors mention (and promise to study in the future), understanding why the variance diminished with time after the interpolation threshold is an important question that could substantially strengthen the paper.*
>
>   We would like to clarify that our focus of this paper is not *understanding why double-descent occurs*. We include the discussion of epoch-wise double descent to motivate the potential of establishing connections between optimization stability of a DNN and its generalization errors, where we found the variance is a good indicator of generalization errors even for a complex error curve like epoch-wise double descent. As shown in our experiments, the proposed OV well indicates the change of generalization errors for cases having epoch-wise double descent. Moreover, epoch-wise double descent is not a necessary condition for applying OV. As shown in Figure 2 (blue curves), we compared the values of OV and generalization errors of DNNs when there are 0% label noises, from which we can see the curves of generalization errors have no epoch-wise double descent, yet the proposed OV still works pretty well. We will add this discussion to the revision.
>
>   While the work of Yang et al. indeed provides an insightful analysis for epoch-wise double descent, we believe our claim is significantly different from that by Yang et al. Specifically, the curve of the variance presented by Yang et al. is a “/\”-shaped curve, whereas in our paper the variance is a “\/\”-shaped curve. The cause of the difference is the choice of the *loss function* used to apply the bias-variance analysis. Unlike previous studies that focus on CE and MSE losses whose variance only presents a “/\”-shaped curve (see Appendix F), we propose to study zero-one loss, which we believe is the best choice to analyze epoch-wise double descent for two reasons. At first, as shown in Figure 9, the variance curve of zero-one loss resembles  a “\/\” shape, and is sufficiently powerful to indicate the variation of generalization errors.  Moreover, as shown in Appendix C, MSE and CE losses can be easily manipulated without changing the decision boundary, and the zero-one loss is more  directly related to the generalization performance.
>
> - *Correlation does not mean much by itself. For example, the train accuracy is very correlated with the test accuracy. There are other empirical quantities that correlate with the test (and provably so at times, see for example https://arxiv.org/pdf/1912.00528.pdf).*
>
>   Training accuracy itself is *not sufficient* to indicate test accuracy, due to issues such as overfitting. And we would like to emphasize the value of OV should be better described by the consistency between the proposed OV and the test accuracy. As shown in Figure 4 and Figure 2, OV changes **consistently** with the test accuracy, even in terms of small fluctuations. As demonstrated in Figure 5, the effectiveness of OV provides potentials including performing early stopping without a validation set, which cannot be achieved using training accuracy.
>
> - *Usually, training for long enough is sufficient to have high correlation with the test. For example in fig 5 we could ask where would lie a model trained for a very long time (say 100 epochs) my bet it would right there on the line of best test error.*
>
>   The mentioned situation does not hold in real scenarios where label noises exist (See our experimental results of SVHN and CIFAR10 in Figure 2).

---

> > ### Comment · AnonReviewer5 · 2020-11-25
> > **Thank you for the response.**
> >
> > I would like to thank the authors for a thoughtful and elaborate response.
> > * I think there is some typo in the response because all shapes are "\/\\" if I'm not mistaken. Anyway, the idea was not the choice of the loss but rather to point out that the variance being the main reason for model Double Descent has already been suggested before and transferring it to epoch double descent is not a big surprise.
> >
> > * I agree that training accuracy is not sufficient by itself albeit it does correlate with the test performance (not perfect correlation of course). But I would argue that OV is not sufficient (or at least I'm not convinced that it is) to indicate test accuracy. If you had to train a model would you use OV to stop the model? My guess is that you would not and you would use a withheld test/validation set. As this has provable guarantees. One indication that OV *won't* have provable guarantees is that it does not correlate across different model (which a validation set of course does). Therefore, more evidence (whether empirical or theoretical) are needed to persuade that we really don't need a test set.
> >
> > * I meant that for large enough models. I agree that for models near the interpolation point training performance is deceitful. But this is not true when the model is large enough, there training for long is usually the right thing to do.
> >
> > * What do you mean the test set is unknown in practice? Of course it is known, we use it every time we do supervised learning. You split the data to train and test (or validation, whichever you prefer to call it) and voila! you have a test set.
> >
> > * While this is not the focus of the paper, my point is that it *should be*, otherwise it is not convincing that it is a fundamental quantity of interest.
> >
> > * While I share the authors' intuition about OV, and could believe that this quantity is interesting to study, right now this is merely an intuition. In order to make the intuition scientific, we need more evidence than mere correlation with test performance across training epochs.
> >
> > Again, I would like to thank the authors for their response and hope that my comments are constructive.

---

> ### Author Response · Authors · 2020-11-18
> **Response to Reviewer 5 - Part 2**
>
> - *If the argument here is early stopping I can just use the test set.*
>
>   The test set is *unknown* in practice.
>
> - *The quantity does not correlate through different model architectures.*
>
>   In fact, OV could correlate through different architectures when they share some similarity, Specifically, as verified in Section 3.4, the values of OV on ResNet18 of different widths well reflect that the size of a network can improve the model’s resilience to sampling noise, which leads to better generalization performance. Improving OV to correlate through model structures with significant variances is not the main focus of this paper, and is left as an important future work.
>
> - *To alleviate my concerns I would love to see some theoretical justification for why is this an important quantity and when does it arise naturally. Alternatively, if we can predict test set across different model architectures, that would give more motivation to as why we might want to look at this quantity more deeply.*
>
>   Thanks for your suggestion, although our paper lacks a rigorous theoretical justification to illustrate the connection between OV and generalization, in this paper we provide a comprehensive study to empirically demonstrate the effectiveness of OV. As shown in our experiments, OV can be a good indicator of the generalization errors without the use of validation set, which we believe may shed the light for many valuable research directions and applications.
>
>   The intuition of OV is quite clear. OV can be regarded as a metric that measures the optimization stability of DNNs with respect to sampling noises, and sampling noises are the major cause of optimization variance -- how drastically the function captured by a DNN varies as the optimization proceeds. Consequently, when the value of OV is large, the generalization error of the DNN is very likely to be bad, since the function learned by DNN has a large variance.
>
>   A similar metric of OV is the sharpness of local minima proposed in [1], which measures the stability of local minima as an indicator of the generalization error. OV can be treated as an extension of this metric, which measures the stability of a DNN during the entire optimization process, instead of just for local minima.
>
>   [1] Keskar et al. On large-batch training for deep learning: Generalization gap and sharp minima. ICLR 2017.
>
> - *Minor Comments: It might be a personal preference, but I find the notation confusing and in disconnect with the standard notation in the field. For example, using t as labels and y as predictions is highly non-standard and would confuse many readers. (why not y for label, \hat y or p for prediction, t for training time?) "The expected loss should be small to ensure good generalization performance" - this is a very informal sentence that I found confusing rather than helpful.*
>
>   Thanks for pointing this out! We will revise our paper to make it clearer and easier to read. However, we should emphasize that the notation follows a previous paper [2] for the convenience of analysis.
>
>   [2] Pedro Domingos. A unified bias-variance decomposition for zero-one and squared loss. AAAI 2000.

---

### Decision · Program_Chairs · 2021-01-07
**Final Decision**

**Decision:**

Reject

**Comment:**

Originality: The paper can be developed into a very nice contribution, if the value of the newly introduced optimization variance is evaluated more thoroughly (e.g., through simple theory, or through more rigorous experiments).

Main pros:
- One of the early works studying epoch-wise double descent
- Optimization variance is an interesting concept and might very well be useful for finding good early stopping points.

Main cons:
- Findings about epoch-wise double descent remain inconclusive
- optimization variance is not sufficiently evaluated to judge its usefulness: theoretical justification for why is this an important quantity and when does it arise naturally is something missing at this point.

Overall: there was a consensus that the paper focuses and provides an interesting story, with new ideas; however, paper's conclusions are not strongly supported by experiments; more experiments are needed to make arguments conclusive.